# Inflammatory cytokines promote interferon regulatory factor (IRF) transcriptional activity in human pulmonary epithelial cells through the induction of IRF1 by nuclear factor-κB

Amandah Necker-Brown[1], Mahmoud M. Mostafa[1], Andrei Georgescu[1], Andrew J. Thorne[1], Priyanka Chandramohan[1], Cora Kooi[1,2], Keerthana Kalyanaraman[1], Alex Gao[1], Akanksha Bansal[1], Sarah K. Sasse[3], Anthony N. Gerber[3,4], Richard Leigh[2], Robert Newton[1]*

**1** Department of Physiology and Pharmacology, University of Calgary, Calgary, Alberta, Canada,
**2** Department of Medicine, Lung Health Research Group, Snyder Institute for Chronic Diseases, Cumming School of Medicine, University of Calgary, Calgary, Alberta, Canada, **3** Department of Medicine, National Jewish Health, Denver, Colorado, United States of America, **4** Department of Immunology and Genomic Medicine, National Jewish Health, Denver, Colorado, United States of America

* rnewton@ucalgary.ca

## Abstract

Interferon regulatory factors (IRFs) play key roles during viral and bacterial infections. However, their regulation by inflammatory cytokines, including interleukin (IL)-1β and tumor necrosis factor (TNF) α, remains underexplored. As airway epithelial cells (AECs) modulate lung inflammation, IRF expression was characterized in pulmonary A549 and bronchial BEAS-2B epithelial cells along with primary AECs grown in submersion, or air-liquid interface, culture. While, IRF6 mRNA was *only* highly expressed in primary cells, IRF4 and IRF8 mRNAs were consistently low across the models. All the other IRF mRNAs were expressed in each model. IRF3 and IRF9 mRNAs were highly expressed, but their proteins remained primarily cytoplasmic post-IL-1β treatment in A549 cells. IRF2 showed moderate/high mRNA expression and was constitutively nuclear. However, RNA silencing did *not* support roles for IRF2 or IRF3, with only a modest role for IRF9, in the IL-1β-induced activation of an IRF reporter. IRF1 mRNA was highly induced by IL-1β in A549 and primary cells. Similarly, IRF1 protein was increased by IL-1β and TNFα in A549 cells, and by TNFα in BEAS-2B cells. In A549 cells, IL-1β-induced IRF1 protein localized to the nucleus and since IRF1 silencing prevented IRF reporter activity, a major transcriptional role was indicated. Mechanistically, the inflammatory transcription factor, nuclear factor (NF)-κB, was necessary for IL-1β- and TNFα-induced IRF1 expression. Further, four novel enhancer regions 5′ to *IRF1* bound the NF-κB subunit, p65, and their IL-1β/TNFα-induced reporter activity required consensus NF-κB motifs. Three such regions recruited RNA polymerase-2 and were flanked by the active chromatin mark, histone 3 lysine 27 acetylation, supporting enhancer involvement in IRF1 transcription.

**Data availability statement:** A549 ChIP-sequencing data are available under GEO accession GSE296100 and GSE296101 and links for the data used in this article are available upon request. BEAS-2B ChIP-seq data are available in GEO under accession GSE79803. A549 RNA-sequencing data are available in Gene Expression Omnibus (GEO) under accession GSE295743. The raw tpm values for the data used in this article are also available from the authors upon request.

**Funding:** This work was supported by: Grants from the Canadian Institutes of Health Research (CIHR) (funding reference numbers: PJT 156310 (RN) & 180480 (RN)) and Natural Sciences and Engineering Research Council of Canada (NSERC) discovery grants [RGPIN-2016-04549 & RGPIN-2023-03763] (RN); studentships (ANB): Alberta Graduate Excellence Scholarships (2020-2025). The funders had no role in study design, data collection and analysis, decision to publish, or preparation of the manuscript. There was no additional external funding received for this study.

**Competing interests:** The authors have declared that no competing interests exist.

Finally, IRF1 expression, transcription rate, and enhancer activity induced by IL-1β, or TNFα, were relatively unaffected by glucocorticoid. IRF1-dependent gene expression may therefore show insensitivity to glucocorticoid and could contribute to glucocorticoid-resistance in diseases that include severe asthma.

## Introduction

During an inflammatory response, cytokines such as interleukin (IL)-1β and tumor necrosis factor (TNF) α bind IL1 and TNF receptors, respectively, on the surface of target cells to activate various signal transduction pathways. This leads to activation of transcription factors, including nuclear factor (NF)-κB, to drive expression of the inflammatory genes that are essential for host defense [1,2]. In each case, signaling from IL1 and TNF receptors activates the IκB-kinases (IKKs), IKKα and IKKβ, which then phosphorylate the cytoplasmic NF-κB inhibitory protein, IκBα [1–4]. This tags IκBα for ubiquitination and rapid degradation to free NF-κB, typically heterodimers of the subunits p50 and p65, which are products of the *NFKB1* and *RELA* genes, respectively. These heterodimers then translocate to the nucleus where they bind NF-κB motifs in the promoter regions of target genes to activate transcription [1,2]. IL-1β and TNFα may also activate transcription factors of the interferon regulatory factor (IRF) family [5–7], which are critical for the expression of genes involved in host defense [8]. IRFs may therefore play additional roles in potentiating inflammatory gene expression. However, most research into the IRFs has focused on their regulation and activation during viral or bacterial infections, where high levels of type 1 and type 2 interferons (IFNs) are believed to promote downstream gene expression [8,9]. Additionally, specific viral and/or bacterial components, termed pathogen associated molecular patterns (PAMPs), are recognized by pattern recognition receptors (PRRs) that also promote activation of IRFs through NF-κB-independent signaling [10,11]. However, some PRRs, such as the family of toll-like-receptors (TLRs), share homology with IL1 receptors and thus can activate NF-κB due to common or overlapping signaling pathways that may also lead to activation of IRFs [12]. Overall, activation of IRFs by NF-κB remains underexplored, and will be a focus for the current study.

Airway epithelial cells (AECs) represent a first target for airborne insults, and, by releasing inflammatory mediators, act as a key driver of inflammatory responses in the lung [13]. Equally, AECs are a primary target for inhaled glucocorticoids, the mainstay anti-inflammatory therapy for asthma [14]. Thus, studying the regulation of transcription factors that potentiate inflammatory gene expression and examining the effect of glucocorticoids on these factors is important to understand how these cells might respond to anti-inflammatory therapy. Although some IRFs are constitutively expressed, others, such as IRF1, are induced by cytokines [8,15], and yet descriptions of IRF expression and their regulation in AEC models are limited [6,16,17]. Indeed, while early analyses of IRF1 promoter function focused on regulation by IFNγ in lymphocyte- and monocyte-derived cell lines [18,19], similar analyses examining

possible regulation by NF-κB in AECs have not been performed. This line of interrogation is of particular importance since IRF1 is known to play roles in the transcriptional regulation of inflammatory genes that include the chemokine, CXCL10, which may be induced by IL-1β or TNFα and is implicated in severe asthma [16,17]. Furthermore, severe, or uncontrolled asthma is associated with viral exacerbations and IRF1 is implicated in these steroid-resistant responses to viral infection [17,20–23]. The current study therefore uses pulmonary epithelial cell lines (A549, BEAS-2B) and primary human bronchial epithelial cells (pHBECs) treated with IL-1β and TNFα to model inflammatory gene expression and signalling in AECs. Additionally, treatment of these cells with glucocorticoids, such as dexamethasone, or the clinically relevant inhaled corticosteroid, budesonide, allows for impacts on inflammatory gene expression to be assessed.

## Results

### Effect of inflammatory cytokines and glucocorticoid on IRF expression in pulmonary cell lines

In A549 cells, maximally effective concentrations for IL-1β (1 ng/ml) and budesonide (300 nM) were previously determined on both IL-1β-upregulated gene expression and in cells harboring NF-κB (6κBtkluc.neo) and glucocorticoid response element (2×GRE.neo.TATA) reporters [24,25]. These concentrations were therefore used to treat A549 cells prior to harvesting for mRNA sequencing (mRNA-seq). This analysis revealed transcripts for all nine IRFs under basal conditions (**Fig 1A and S1A in S1 File**). IRF3 was most highly expressed at 87.2–108.2 transcripts per million (tpm), followed by IRF9, IRF2, and IRF1, which ranged between 7.23–16.2 tpm. IRF5 and IRF7 showed 6.24–9.11 tpm and 1.22–3.65 tpm, respectively (**Fig 1A**). More modestly expressed transcripts included IRF6 (0.06–0.69 tpm) and IRF8 (0.03–0.51 tpm) (**S1A Fig in S1 File**). IRF4 (0.00–0.23 tpm) had an average tpm below 0.01, a cutoff commonly used to filter out lowly expressed genes [26], where functional relevance is thought less likely (**S1A Fig in S1 File**). mRNA-seq performed on untreated BEAS-2B cells showed that relative basal mRNA expression levels of all nine IRFs were similar to A549 cells, with IRF3 being the most highly expressed (87.5–107.6 tpm), followed by IRF9, IRF1, IRF7 and IRF2 (all ranged between 11.5 and 49.8 tpm) (**S1B Fig in S1 File**). IRF6, IRF8 and IRF4 were also lowly expressed (below 0.87 tpm) in BEAS-2B cells (**S1B Fig in S1 File**).

The mRNA-seq data in A549 cells showed IRF1 to be highly upregulated at all times by IL-1β (all $P \leq 0.001$), with maximal expression achieved at 2 h (~40-fold over basal levels) (**Fig 1A**). At 1, 2, 6 and 12 h, budesonide alone showed little effect on IRF1 mRNA, while the addition of IL-1β withbudesonide decreased IL-1β-induced IRF1 mRNA ($P \leq 0.01$) by 25.1–53.8%, depending on time point. In contrast, IRF3 mRNA was minimally impacted by each treatment, with the greatest effect apparent at 6 h, where the combination treatment decreased basal expression by 12.0% ($P \leq 0.05$) (**Fig 1A**). IRF2 was also minimally impacted by each treatment but was upregulated 1.9-fold ($P \leq 0.001$) by IL-1β at 6 h, with a similar effect also apparent for IL-1β plus budesonide (**Fig 1A**). At 24 h, IRF2 was lowly upregulated (1.7-fold, $P \leq 0.01$) by the combination of budesonide and IL-1β, with a similar induction by budesonide alone ($P \leq 0.01$). IL-1β decreased IRF5 mRNA by 0.6-fold at 12 h ($P \leq 0.001$) and 0.7-fold at 24 h ($P \leq 0.05$) and budesonide was largely without effect. At 6, 12 and 24 h, IRF7 and IRF9 mRNAs were both upregulated by IL-1β (all $P \leq 0.01$). In each case, maximal mRNA expression occurred at 6 h with 20.9-fold inducibility for IRF7 and 5.5-fold for IRF9 prior to steadily reducing thereafter. Between 6–24 h, IL-1β-induced IRF7 and IRF9 mRNAs were repressed to near basal levels by the co-addition of budesonide ($P \leq 0.01$), which showed little effect on its own (**Fig 1A**). IRF4 and IRF6 mRNAs remained low with all treatments (**S1C Fig in S1 File**). At 2 h, IRF8 mRNA expression was increased to 0.75 tpm by IL-1β and 0.42 tpm by IL-1β plus budesonide but was otherwise unaltered by any treatment at other times (**S1C Fig in S1 File**).

In A549 cells, IRF1 protein was expressed at low levels in unstimulated cells but following IL-1β treatment was upregulated 415-fold ($P \leq 0.001$) at 2 h and 200-fold ($P \leq 0.001$) at 4 h (**Fig 1B**). By 6 h, IL-1β-induced IRF1 protein expression was much reduced and remained low for up to 24 h. Similar expression kinetics were also apparent following TNFα stimulation, where peak IRF1 protein expression was observed at 2 h (**S2 Fig in S1 File**). Expression of IRF2 and IRF3

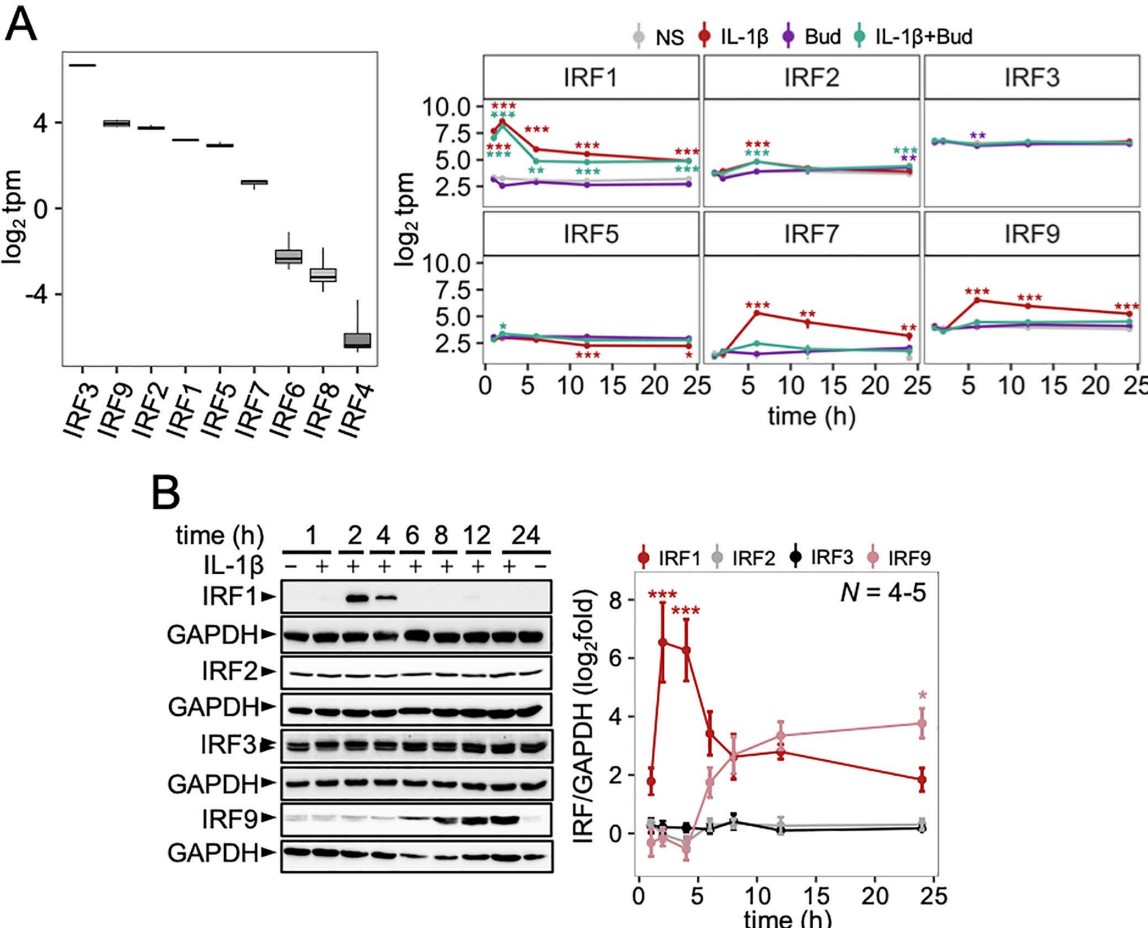

**Fig 1. IRF expression in A549 cells following stimulation with IL-1β.** A549 cells were either not stimulated (NS) or treated with IL-1β (1 ng/ml) and/or budesonide (300 nM; Bud) for the indicated times. (A) RNA from 4 independent experiments was prepared prior to RNA sequencing and results for unstimulated cells (left panel) or following treatments for the indicated times (right panel) are presented as $\log_2$ transcripts/million (tpm). (B) Following either no stimulation or treatment with IL-1β (1ng/ml) for the indicated times, cells from 4–5 independent experiments (N) were harvested for total protein and western blot analysis. All data are plotted mean±SE or box-and-whisker plots. Using tpm or normalized IRF/GAPDH values, significance was tested by one-way ANOVA with Tukey's post-hoc test. * $P \leq 0.05$, ** $P \leq 0.01$, *** $P \leq 0.001$ indicates significance relative to NS.

proteins were unchanged by IL-1β stimulation (**Fig 1B**). IRF3 protein expression was also unchanged by TNFα stimulation (**S2 Fig in S1 File**). While analysis of IRF9 protein revealed upregulation by IL-1β from 6 h onward with a maximal 16.8-fold at 24 h (Fig 1B), western blotting for IRF7 failed to detect a signal (not shown). The four most highly expressed IRFs in basal conditions (IRF1, 2, 3 and 9), for which available antibodies were validated (**Fig 2**), were selected for further functional interrogation.

## IRF roles in transcriptional action by IL-1β

A549 cells were stably transfected with a reporter plasmid containing 6 tandem repeats of an IRF enhancer sequence upstream to a basal promoter driving a luciferase gene. IRF reporter activity was concentration-dependently induced by IL-1β ($EC_{50}$ = 0.07 ng/ml) (**S3A Fig in S1 File**) and maximal luciferase activity occurred 8 h post IL-1β stimulation (13.8 ± 2.4-fold, $P \leq 0.05$) (**S3B Fig in S1 File**).

IRF reporter cells were treated with transfection lipid, increasing concentrations of a control siRNA pool, or siRNA pools targeting each of IRF1, 2, 3 or 9 (**Fig 2A**). Western blotting for IRF1 and 9 following IL-1β treatment for 2 or 24 h (respectively) also confirmed a concentration-dependent loss of the respective protein by the targeting siRNA pools, but not the control siRNA pool (**Fig 2A**). Similarly, unstimulated cells for the constitutive IRFs, IRF2 and 3, confirmed a concentration-dependent knock-down of their protein expression by the targeting, but not control, siRNA pool (**Fig 2A**). For IRF1, the 76.1% knockdown of IL-1β-upregulated protein achieved at 10 nM siRNA resulted in a near complete ablation ($P \leq 0.001$) of IL-1β-induced IRF reporter activity. As similar effects were also achieved with each of the four individual IRF1-targeting siRNAs that made up the pool, inhibition due to selective IRF1-targeting is supported (**S3C Fig in S1 File**). Conversely, the near complete (94.3%, $P \leq 0.05$) knockdown of IRF2 protein at 1 nM siRNA showed little effect on IL-1β-upregulated IRF reporter activity (**Fig 2A**). At 10 nM, the IRF2 siRNA pool produced a minor, but significant, decrease in IL-1β-activated IRF reporter activity (**Fig 2A**). However, since IRF2 protein knockdown was not any greater at 10 nM than at 1 nM, this decrease in luciferase activity is likely due to an off-target effect. Significant 79.1 and 86.5% decreases in IRF3 protein were achieved with 1 and 10 nM siRNA, but these showed no effect on IL-1β-induced IRF reporter activity (**Fig 2A**). At 0.01–10 nM, the IRF9 siRNA pool produced 50.79–99.00% (all $P \leq 0.01$) knockdowns of IRF9 protein and, for concentrations over 0.1 nM, this resulted in 43.1–48.8% (all $P \leq 0.05$) inhibitory effects on IL-1β-induced IRF reporter activity (**Fig 2A**).

To explore the presence of IRF proteins in the nucleus, A549 cells were treated with IL-1β prior to fractionation into cytoplasmic and nuclear extracts and western blotting. IL-1β-induced expression of IRF1 protein was maximally detected at 2 h in the nuclear extracts with no, or very low, expression apparent in the cytoplasmic extracts (**Fig 2B**). Nuclear IRF1 expression was markedly reduced by 4 h but remained detectible at 6 h. The constitutively expressed IRF2 was only detected in the nucleus and this was unchanged by IL-1β (**Fig 2B**). By contrast, IRF3, which was also constitutively expressed, was only present in cytoplasmic fractions, even following 0.5–6 h of IL-1β stimulation (**Fig 2B**). While IRF9 expression was low and remained cytoplasmic up to 4 h post-IL-1β stimulation, by 6 h, a time where total IRF9 expression first increased, IRF9 presence was faintly detected in the nucleus (**Fig 2B**). Taken with the IRF reporter data, no clear role for IRF3 or, despite nuclear localization, IRF2, was shown. Similarly, evidence for activation of IRF9 by IL-1β was weak prior to 6 h and is consistent with the lesser effects on IRF-reporter activity. Conversely, IRF1 showed robust IL-1β-induced nuclear localization and was necessary for IL-1β-stimulated IRF transcription.

### Effect of inflammatory cytokines and glucocorticoid on IRF1 expression in pulmonary cell lines

In A549 cells, the induction of IRF1 protein expression at 2 h by IL-1β and TNFα was concentration-dependent with maximal effects reached at 0.1–1 ng/ml and 1–10 ng/ml with $EC_{50}$ values of 0.03 ng/ml (95% confidence interval: 0.01–0.3) and 0.8 ng/ml (95% confidence interval: 0.4–1.5), respectively (**Fig 3A**). At this time, dexamethasone co-treatment with IL-1β caused a modest (36.5%), but not significant, reduction in IRF1 protein (**Fig 3B**). A similar moderate reduction (17.4%, $P \leq 0.05$) in IRF1 protein by dexamethasone was apparent for TNFα-induced IRF1 (**Fig 3B**). At 6 h, dexamethasone produced a significant, but again modest (34.1%, $P \leq 0.05$) reduction in IL-1β-induced IRF1 protein, with a non-significant 7.8% reduction in TNFα-induced IRF1 (**Fig 3B**). In BEAS-2B cells, TNFα strongly induced IRF1 protein expression at 2 h but this was unaffected by dexamethasone co-treatment (**Fig 3C**). By 6 h, TNFα-induced IRF1 was still strongly detected, but repression in the context of dexamethasone co-treatment had further increased to 56.9% ($P \leq 0.01$).

### NF-κB-dependent transcriptional control of IL-1β- and TNFα-induced IRF1 expression

To test for transcriptional control of IRF1 expression, qPCR was initially used to confirm a rapid, by 1 h, upregulation of IRF1 mRNA by IL-1β (**S4A Fig in S1 File**). This peaked at 2 h ($P \leq 0.05$) and remained elevated at 4 h prior to declining to lower steady-state levels from 6 h onwards. These data also revealed little or no repression at any time by

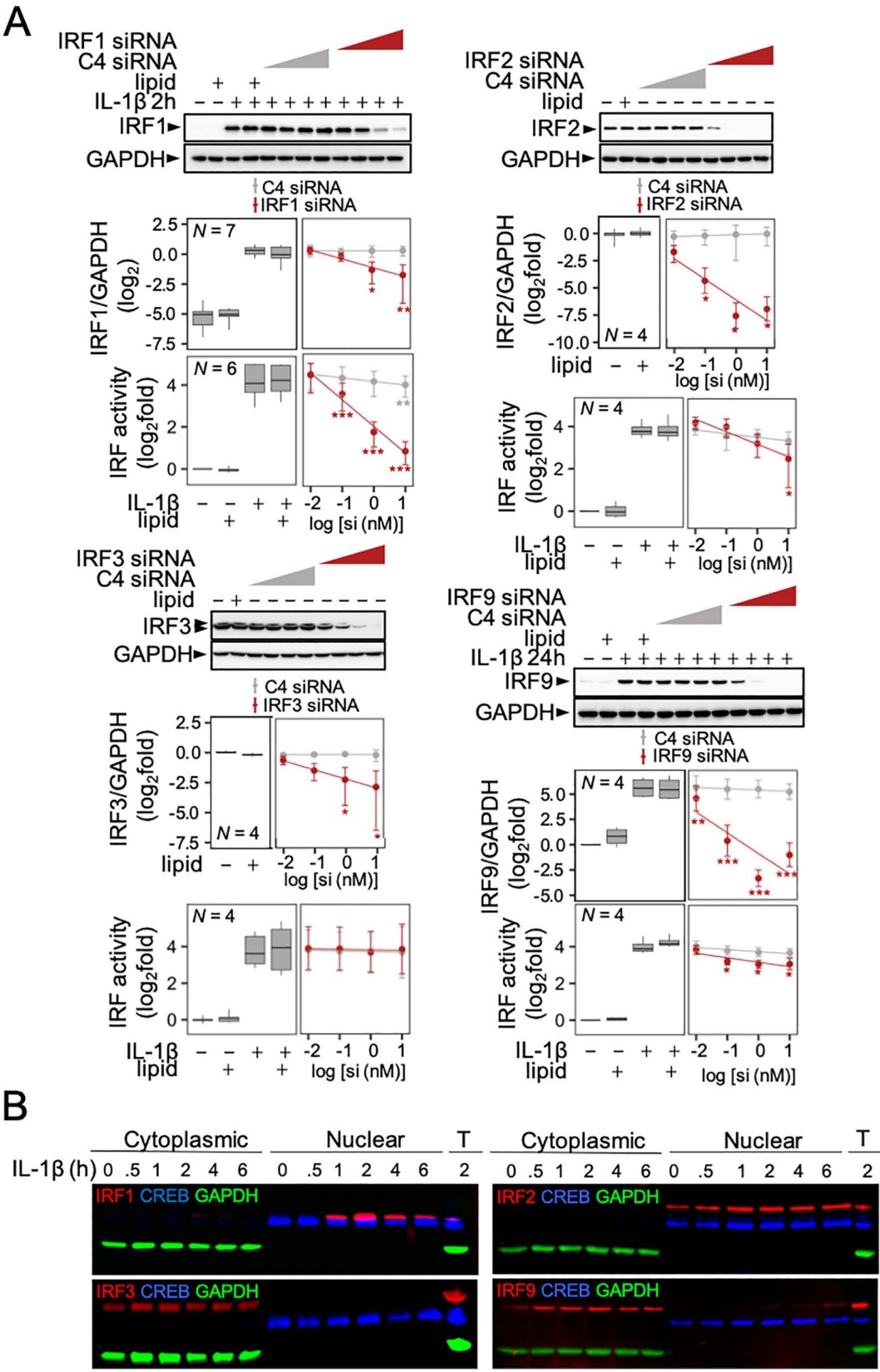

**Fig 2. IRF translocation and transcriptional activity following treatment with IL-1β.** (A) A549 cells stably transfected with an IRF luciferase reporter were treated with increasing concentrations (0.01, 0.1, 1 or 10 nM) of; control siRNA pool (C4), the indicated IRF siRNA pool, or a lipid control (lipid) prior to either no stimulation or treatment with IL-1β (1 ng/ml). Cells were harvested at 0 h (i.e., no stimulation) for IRF2 and IRF3 protein or at 2 h for IRF1 protein, 24 h for IRF9 protein, and 8 h for luciferase determination. Blots representative of $N$ independent experiments are shown. Following normalization to GAPDH, western blot data were plotted as $\log_2$ IRF/GAPDH. IRF reporter data are plotted as $\log_2$ fold of no stimulation. All data are shown as either mean ± SE or box-and-whisker plots. Using normalized IRF/GAPDH values or relative light units, significance was tested by one-way ANOVA with Tukey's post-hoc test. * $P \le 0.05$, ** $P \le 0.01$ or *** $P \le 0.001$ indicates significance relative to IL-1β treated cells (IRF1, IRF9) or non-stimulated cells (IRF2, IRF3). (B) A549 cells were treated with IL-1β (1 ng/ml) at the indicated times prior to fractionation into 'Cytoplasmic' and 'Nuclear' lysates, or harvested for total protein (T), prior to western blotting using fluorescently tagged antibodies. Images are representative of 3 or more fluorescent, or equivalent chemiluminescent, blots.

dexamethasone as a cotreatment. Subsequently, qPCR was used to detect unspliced nuclear IRF1 RNA as a surrogate of transcription rate. This demonstrated a peak in unspliced IRF1 RNA at 30 mins to 1 h post-IL-1β that then declined steeply from 1–6 h. There was also no material effect of dexamethasone cotreatment with IL-1β at either 1 or 4 h on unspliced IRF1 RNA (**S4B Fig in S1 File**). These data support rapid activation of IRF1 gene transcription by IL-1β in a manner that was largely unaltered by glucocorticoid co-treatment.

Adenoviral overexpression of a dominant inhibitor of NF-κB, IκBαΔN, was used to test the role of NF-κB in the induction of IRF1 [27]. At a multiplicity of infection (MOI) of 30, Ad5-IκBαΔN maximally inhibits NF-κB-dependent transcription induced by IL-1β [24,28]. At this MOI, Ad5-IκBαΔN or as a control, Ad5-GFP, produced robust expression of the truncated IκBα protein or GFP, respectively (**Fig 4A**). When stimulated for 2 h with IL-1β or IL-1β plus budesonide, the induction of IRF1 protein expression was ablated by Ad5-IκBαΔN ($P \le 0.05$), but unaffected by Ad5-GFP. IL-1β-inducibility of IRF1 mRNA was also prevented by Ad5-IκBαΔN when compared to Ad5-GFP ($P \le 0.001$) (**Fig 4B**).

A pool of p65-targeting siRNAs was previously shown to concentration-dependently reduce p65 protein expression and attenuate NF-κB reporter activity when activated by IL-1β [24]. Using 1 nM of this p65 siRNA pool, constitutive p65 expression and IRF1 protein expression induced by IL-1β were reduced 61.4% ($P \le 0.005$) and 91.9% ($P \le 0.001$) respectively compared to no siRNA (naïve) control (**Fig 5A**). Similarly, increased IRF1 mRNA expression induced by IL-1β or TNFα, each in the presence or absence of dexamethasone, were all reduced ($P \le 0.05$) by p65 siRNA compared to the no siRNA (naïve) control (92.9% for IL-1β or 92.4% for TNFα) (**Fig 5B**). Control siRNA pool at 1 nM had no impact compared to naïve conditions on either IRF1 mRNA or protein.

The small molecule IKKβ inhibitors, PS-1145, ML-120B and TPCA-1 all reduced IL-1β-induced IRF1 mRNA expression with $EC_{50}$ values in the 1–10 μM range and in each case 30 μM was confirmed as showing near-maximal inhibition of IRF1 mRNA expression (**Fig 6A**). As these $EC_{50}$ values were similar to those reported on a stably transfected NF-κB reporter, as well as other NF-κB dependent genes, the data are consistent with repression of IRF1 mRNA also being due to NF-κB inhibition [24,29]. Using 30 μM of each IKKβ inhibitor, IRF1 protein induced by both IL-1β and TNFα was also reduced (**Fig 6B**). While PS-1145 and ML-120B resulted in 36.8% and 43.2% repression of IL-1β induced IRF1 ($P \le 0.05$), respectively, TPCA-1 produced a more profound repression (97.8%, $P \le 0.001$). Very similar levels of repression were observed with TNFα-induced IRF1 (**Fig 6B**).

## The NF-κB subunit p65 binds the *IRF1* promoter

Recruitment of p65 and RNA polymerase 2 (POL2) to the *IRF1* locus was examined using chromatin immunoprecipitation (ChIP) data from A549 and BEAS-2B cells that had been stimulated for 1 h with IL-1β (1 ng/ml) or TNFα (20 ng/ml), without or with budesonide (300 nM) or dexamethasone (100 nM), respectively [30]. The A549 data revealed 3 main regions (R1, R2, R3) showing IL-1β-induced recruitment of p65 within ~10 kb upstream to the *IRF1* gene transcription start site (TSS) (**Fig 7A**). An additional 2 regions (R2.2, R4) with lower p65 recruitment were also apparent. In BEAS-2B cells, TNFα induced the greatest p65 recruitment to R1, with more minor levels of recruitment to the remaining four regions (R2,

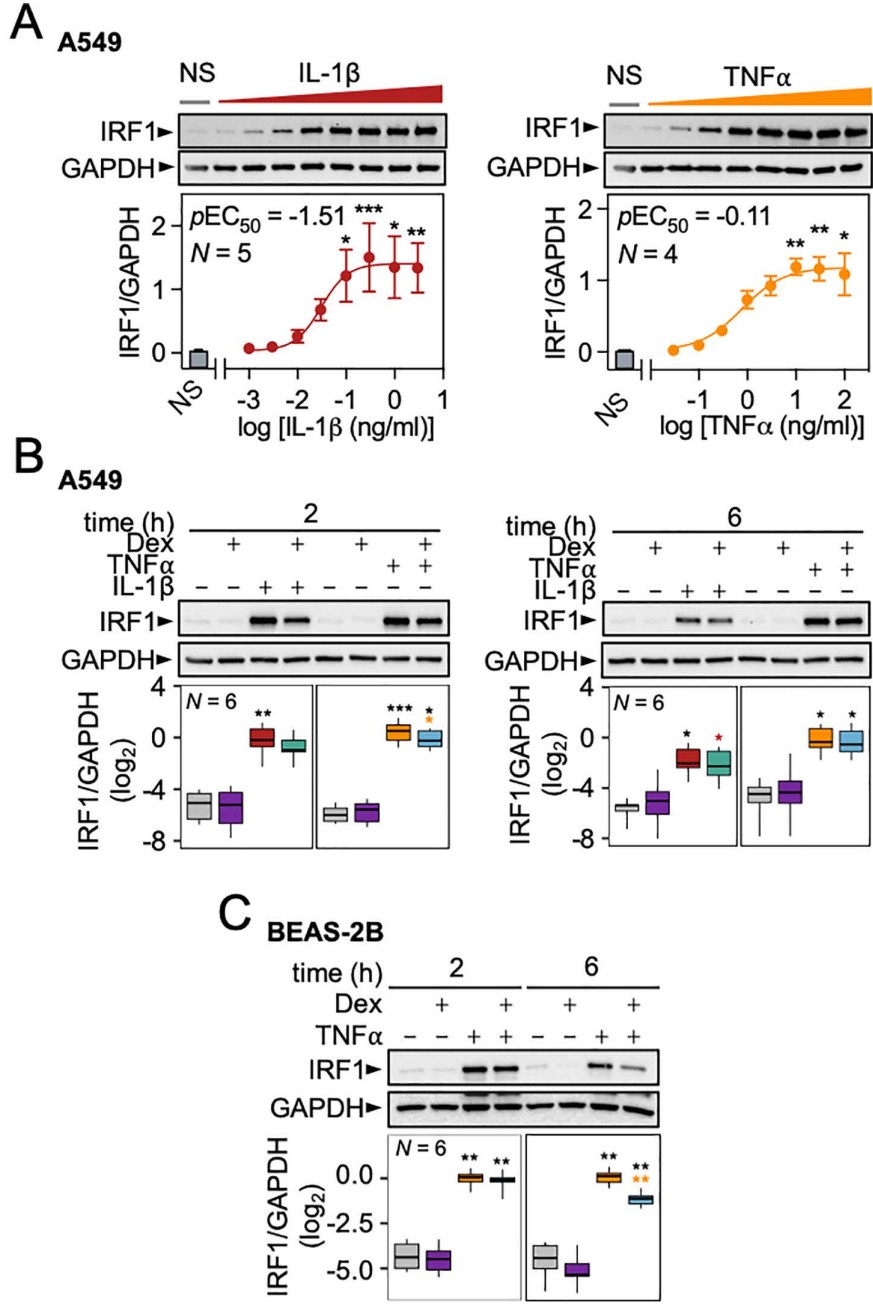

**Fig 3. IRF1 protein expression in airway epithelial cells.** (A) A549 cells were either not stimulated (NS) or treated with 0.003, 0.01, 0.03, 0.1, 0.3, 1, 3 and 10 ng/ml of IL-1β or 0.03, 0.1, 0.3, 1, 3, 10, 30 and 100 ng/ml TNFα for 2 h before harvesting for western blot analysis. (B) A549 or (C) BEAS-2B cells were either NS or treated with IL-1β (1 ng/ml), TNFα (10 ng/ml) and/or dexamethasone (1 μM; Dex) for 2 or 6 h as indicated prior to harvesting for western blot analysis. Data from $N$ independent experiments are plotted mean ± SE or box-and-whisker plots. Using normalized IRF1/GAPDH values significance was tested by one-way ANOVA with Tukey's post-hoc test. * $P \leq 0.05$, ** $P \leq 0.01$, *** $P \leq 0.001$ indicates significance relative to NS cells (black), or IL-1β/TNFα stimulated cells as indicated in red and yellow respectively.

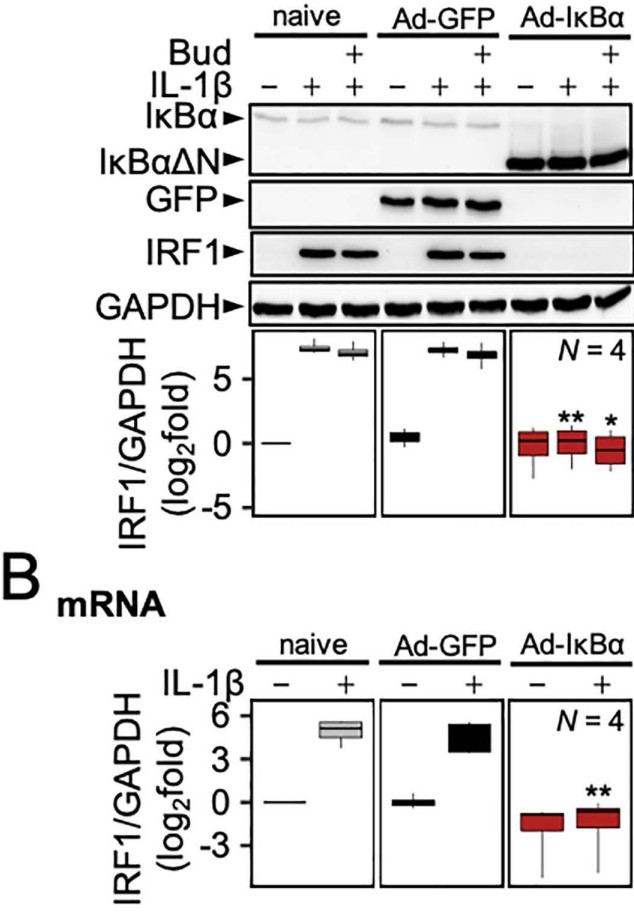

**Fig 4. IκBαΔN overexpression prevents IL-1β-induced IRF1 expression.** A549 cells were infected with Ad-IκBαΔN or control Ad-GFP at a MOI of 30 for 24 h prior to treatment with IL-1β (1 ng/ml) and/or budesonide (300 nM), as indicated. (A) Total protein was harvested at 2 h for western blot analysis of GFP, IκBαΔN, IRF1 and GAPDH. Representative blots are shown and data from *N* independent experiments are plotted log₂ fold (IRF1/GAPDH). (B) Cells were harvested at 2 h and RNA was extracted for qPCR analysis of IRF1 and GAPDH mRNA. Following normalization to GAPDH, data were expressed as log₂ fold of untreated. All data are shown as box-and-whisker plots. Using normalized IRF1/GAPDH values, significance was tested by one-way ANOVA with a Tukey's post-hoc test. * $P \leq 0.05$, *** $P \leq 0.001$ indicates significance relative to the no-virus (naïve) control within each treatment group (not-stimulated, IL-1β, or IL-1β-Bud treated).

R2.2, R3 & R4). TNFα also appeared to induce p65 recruitment to a region that flanked R1. In A549 cells, IL-1β increased POL2 recruitment to R4, with more moderate recruitment to R1, as well as to the 3′ end of the *IRF1* locus, where no p65 enrichment was observed in untreated cells (**Fig 7A**). Similarly, in BEAS-2B cells, TNFα treatment increased recruitment of POL2 to R4 and the 3′ end of the *IRF1* gene. In addition, POL2 recruitment to the R1 region and just upstream to R2 was also observed. While in A549 cells, there appeared to be a minor reduction in all the p65 enrichment peaks following the addition of glucocorticoid to IL-1β, there was no clear change in POL2 recruitment. In BEAS-2B cells, TNFα-induced recruitment of p65 to the R1-R2.2 regions was not impacted by glucocorticoid co-addition. Additionally, with the exception of R4, there was a reduction in the overall TNFα-induced recruitment of POL2 along the entire *IRF1* locus in the presence of glucocorticoid. Since, in A549 cells, *IRF1* was the only IL-1β-upregulated gene within ~200kb of these IL-1β-induced

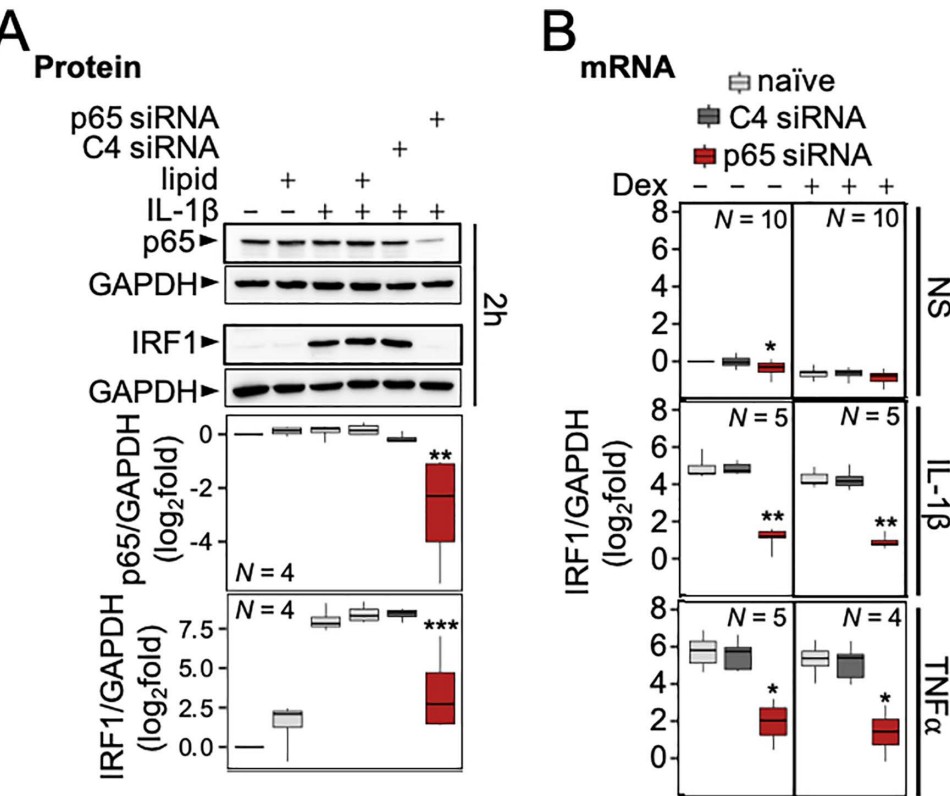

**Fig 5. The NF-κB subunit, p65, is necessary for IL-1β and TNFα-induced IRF1 induction.** (A) A549 cells were either not treated or treated with lipid or lipid in the presence of 1 nM of control siRNA pool (C4) or p65 siRNA pool. After stimulation with IL-1β (1 ng/ml), total protein was harvested at 2 h for western blot analysis. Representative blots are shown. Following normalization to GAPDH, data were plotted as log$_2$ fold over not-stimulated (B) A549 cells were treated with 1 nM of each siRNA pool as in A prior to treatment with IL-1β (1 ng/ml) or TNFα (10 ng/ml) in the absence or presence of dexamethasone (1 μM, Dex), as indicated. Total RNA was extracted after 2 h for qPCR analysis of IRF1 and GAPDH. All data are shown as box-and-whisker plots. Using normalized IRF1/GAPDH values, significance was tested by one-way ANOVA with a Tukey's post-hoc test. * $P \leq 0.05$, ** $P \leq 0.01$, *** $P \leq 0.001$ indicates significance relative to IL-1β-stimulated cells within each siRNA group (naïve, C4 or p65).

p65 ChIP-seq peaks (**Fig 7B**), these regions (R1 – R4) are consistent with transcriptional control of the *IRF1* gene in A549 cells.

Examining ~10 kb upstream of the *IRF1* TSS in the UCSC genome browser identified 10 robust p65 or c-Rel motifs that were defined by confidence scores of ≥ 400 (i.e., $P \leq 10^{-4}$) for the position-weight matrices in the JASPAR database and are collectively referred to as NF-κB motifs (**Fig 7C** and **D**) [31,32]. Six of these motifs were located centrally to the p65 ChIP-seq peaks spanning regions R1, R2, R3 and R4. These regions, specifically the NF-κB motifs, also aligned with the most conserved sequences amongst a group of 100 vertebrates and is therefore strongly suggestive of key functional roles (**Fig 7D**) [31,33]. While R2.2 showed areas of high conservation, there were no strong NF-κB motifs. However, the presence of a STAT1 motif, which may be active in the context of IFNγ stimulation [18,19], may account for the high level of conservation apparent within R2.2. Consistent with little or no regulation of IRF1 expression by glucocorticoids, the only GR site found within the ~10 kb *IRF1* promoter did not align to a conserved region and thus may not play an important regulatory role. Along with genetic conservation, acetylation of histone 3 at lysine 27 (H3K27) associates with open/active chromatin regions [34–37]. Publicly available acetylated H3K27 ChIP-seq data from A549 cells (obtained from the ENCODE portal with the identifier: ENCFF831UJW) revealed that this mark flanked R1, R2, R2.2 and R4, but not the R3 region (**Fig 7D**) [38,39].

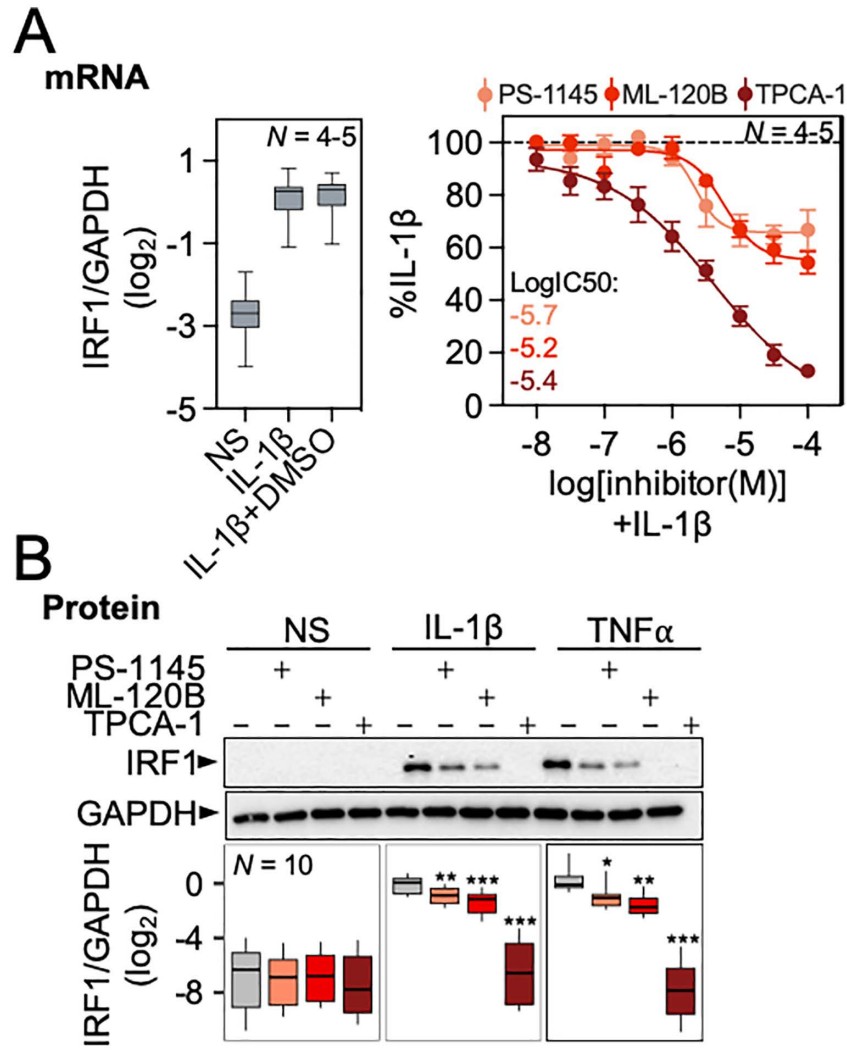

**Fig 6. Inhibitors of IKKβ reduce the upregulation of IRF1 by IL-1β and TNFα.** A549 cells were either incubated with DMSO at a final concentration of 0.3%, or incubated with TPCA-1, ML-120B or PS-1145, at (**A**) the indicated concentrations, or at (**B**) 30 μM, for 1.5 h prior to no stimulation (NS) or treatment with TNFα (10 ng/ml) or IL-1β (1 ng/ml) as indicated. After 2 h, cells were harvested for qPCR and western blot analysis of IRF1 and GAPDH. Data from $N$ independent experiments were normalized to GAPDH and expressed as $\log_2$ fold of untreated or as a % of IL-1β-treated. Data are plotted as box-and-whisker plots or mean ± SE. Using normalized gene/GAPDH values, significance was tested by one-way ANOVA with a Tukey's post-hoc test. * $P \leq 0.05$, ** $P \leq 0.01$, *** $P \leq 0.001$ indicates significance relative to the no-inhibitor (naïve) control within each treatment group (NS, IL-1β or TNFα).

### Four regions in the *IRF1* promoter drive IL-1β-induced transcription

To explore the ability of different genomic regions to drive transcription, each region, as depicted in Fig 7D, was cloned into a luciferase reporter plasmid. The proximal promoter regions, designated PP.1 and PP.2, each contained the *IRF1* TSS and either R4, or both R3 and R4, respectively. The more distal promoter (DP) region located ~6 kb upstream to the TSS was cloned as DP.1–3 and contained regions R1 – R2.2, R2 and R2.2 or just R2.2, respectively (**Fig 7D**). Following stable transfection into A549 cells, the greatest reporter drive under basal conditions was found for DP.3, which contains R2.2, whereas DP.2, containing both R2 and R2.2, revealed a much reduced reporter activity that suggests presence of repressive elements within this region (**S5A Fig in S1 File**). DP.1 and both the proximal promoter clones, PP.1 and PP.2, showed basal activity that was intermediate to DP.3 and DP.2.

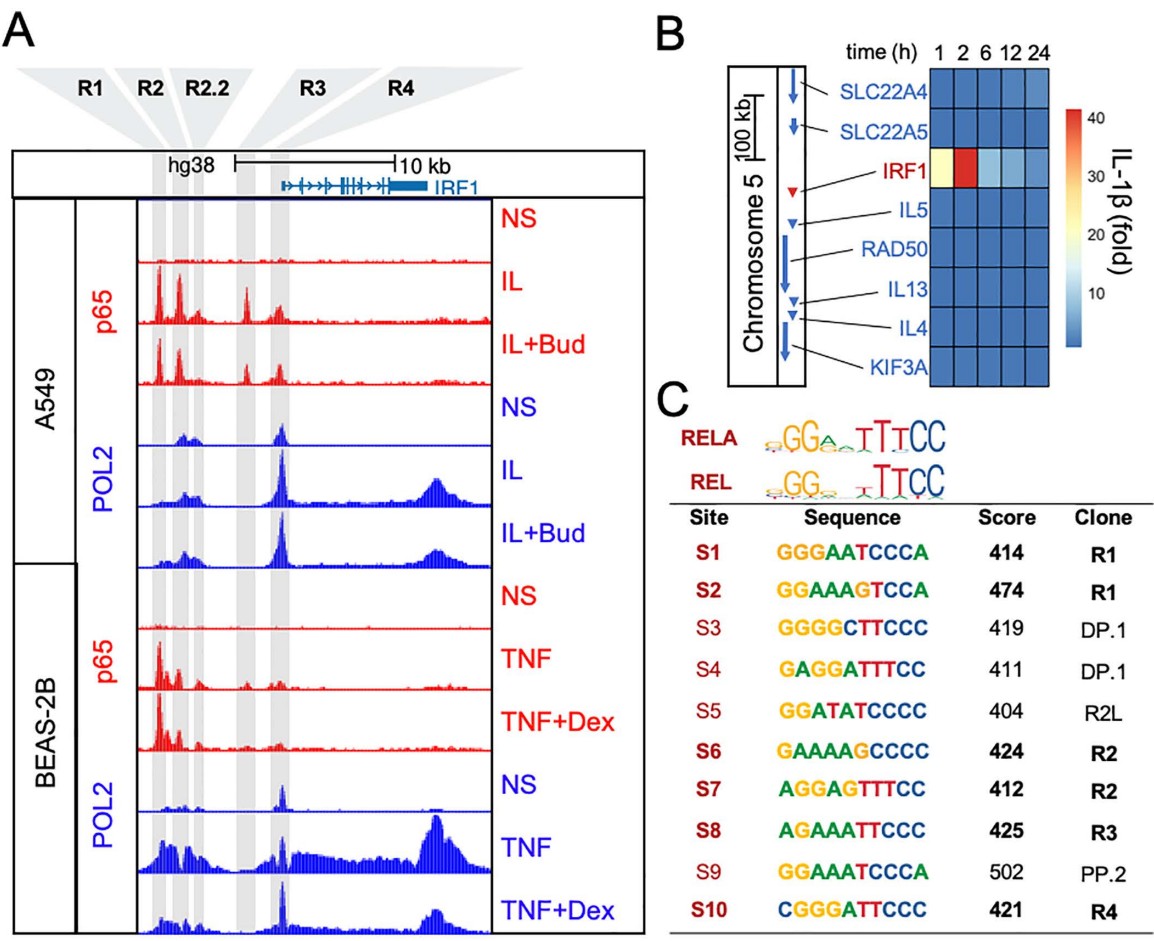

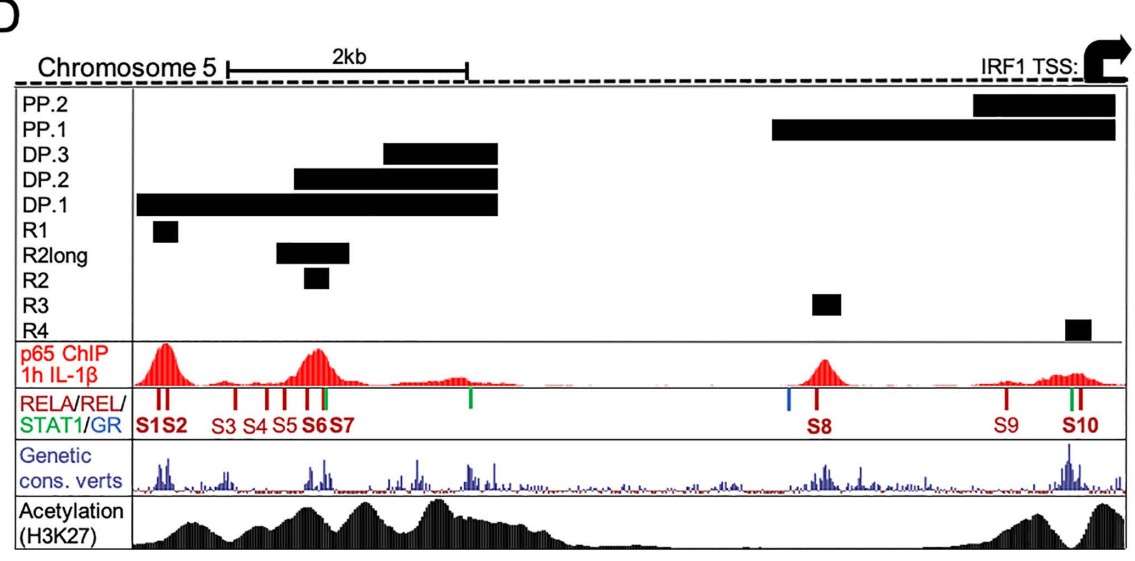

**Fig 7. Recruitment of p65 and RNA polymerase 2 to the *IRF1* gene promoter.** (A) A549 cells were either not stimulated (NS) or stimulated with IL-1β (1 ng/ml) or IL-1β + budesonide (300 nM, Bud) for 1 h prior to ChIP-sequencing. BEAS-2B cells were either not stimulated or treated with TNFα (20 ng/ml) or TNFα + dexamethasone (100 nM, Dex) for 1 h prior to ChIP-sequencing (data from Kadiyala et al., 2016). Peak heights reflect normalized factor occupancy for p65 and RNA polymerase 2 (POL2) and are visualized for the *IRF1* locus using the UCSC Genome Browser depicting the five locations that bound p65 (R1-R4) which were interrogated in Fig 8. (B) Heatmap showing mRNA sequencing data, expressed as average fold of NS at each timepoint, for genes within 200 kb of the *IRF1* gene locus following IL-1B (1 ng/ml) treatment of A549 cells for the indicated times. Gene loci are shown to the left and correspond (top to bottom) with genes in the heatmap. (C) Frequency matrix logos for RELA and REL from the transcription factor binding database JASPAR are shown with the sequences and probability scores for the 10 RELA/REL JASPAR motifs upstream to *IRF1*. (D) Schematic upstream of the *IRF1* locus depicting the regions that were cloned into luciferase plasmids in Fig 8 (DP: Distal Promoter, PP: Proximal Promoter, R: short enhancer region) as black bars above the p65 ChIP sequencing track. The first track shows p65 ChIP sequencing in A549 cells after 1 h IL-1β treatment. On the second track, numbered red bars represent REL and RELA motifs with JASPAR with scores ≥ 400 (as in C) while blue and green bar indicate STAT1 and GR motifs. The third track shows genetic conservation amongst a group of 100 vertebrates as obtained from the UCSC Genome Browser Database (31). The fourth track shows ChIP sequencing of H3K27 in unstimulated A549 cells (ENCODE portal identifier: ENCFF831UJW, Tim Reddy Laboratory). All tracks visualized on the UCSC genome browser.

Following IL-1β treatment, the two proximal promoter constructs, PP.1 and PP.2, revealed reproducible (both $P \leq 0.001$), but modest (less than 2-fold) inducibility that was further reduced by dexamethasone (Fig 8A). In each case, dexamethasone alone reduced the basal reporter drive by 13.0–18.3%, an effect that was significant for PP.1. While the R1-R2.2-containing construct, DP.1, was not upregulated by IL-1β, the two shorter constructs, DP.2 and DP.3, each revealed significant, but again, at 1.6- and 1.4-fold, very modest inducibility by IL-1β (Fig 8A). In the presence of dexamethasone, DP.1 and DP.3 revealed reductions in reporter activity that were also apparent with IL-1β plus dexamethasone relative to IL-1β alone. DP.2 was not affected by dexamethasone, whether alone or in combination with IL-1β. Since transfection of the parent reporter plasmid into A549 cells does not show IL-1β-induced responsiveness [29], the above data are generally suggestive of IL-1β responsive DNA regions in both the proximal and distal regions. However, in each case the two shorter fragments, cloned into PP.2 and DP.3, produced only modest, 1.3 and 1.4-fold increases, respectively, in reporter activity that were further reduced in the longer proximal PP.1 and distal DP.2 and DP.3 constructs. While these data indicate only low reporter activity for PP.2 and DP.3, which contain the weak p65-binding regions, R4 and R2.2, respectively, the fact that regions containing the robust p65-binding regions, R1, R2 and R3, did not induce substantial reporter activity was unexpected. However, since increased distance from motif to TSS is known to adversely affect reporter activity [40], these data are perhaps less surprising.

To specifically test transcriptional drive from the four main p65-binding regions, shorter 202–236 bp regions, spanning each peak of p65 enrichment, were cloned into the basal luciferase reporter. In addition, a 617 bp region (R2long), containing the NF-κB motif, S5, as well as the S6 and S7 motifs that were more central to the p65 peak, was also cloned into the basal reporter.

These R1-R4 reporters all revealed similar levels of basal luciferase activity (**S5B Fig in S1 File**). Following IL-1β-stimulation, inducibility was apparent for all constructs with similar effects evident in both the forward and reverse directions (**Fig 8B**). The R1 reporter, with the region showing the highest p65 enrichment peak, resulted in the greatest overall fold inducibility (19.1-fold, $P \leq 0.001$) in the forward orientation. While this response was variable in the presence of dexamethasone, there was no net effect and this is consistent with equivalent effects observed for the reverse (R1R) construct. For the R2 and R2long reporters, matched analyses revealed similar 4.8–6.6-fold inducibility for the short and long constructs in both the forward and reverse orientations (**S5B and C Fig in S1 File**). This suggests all the main elements responsible for inducibility were present within the shorter R2 construct. Further, dexamethasone alone had no effect on these four R2 constructs, but in each case resulted in some loss (11.1% to a more variable 44.5% for R1) of IL-1β-induced reporter activity (**Fig 8B**). The forward and reverse R3 clones both revealed similar 4.6–4.7-fold inducibility by IL-1β (**Fig 8B**). Dexamethasone alone showed no effect on these R3 reporters, but in the presence of IL-1β resulted in a modest (18.3%, $P \leq 0.001$) reduction in activity for the forward construct while the reverse construct was increased from 4.7- with

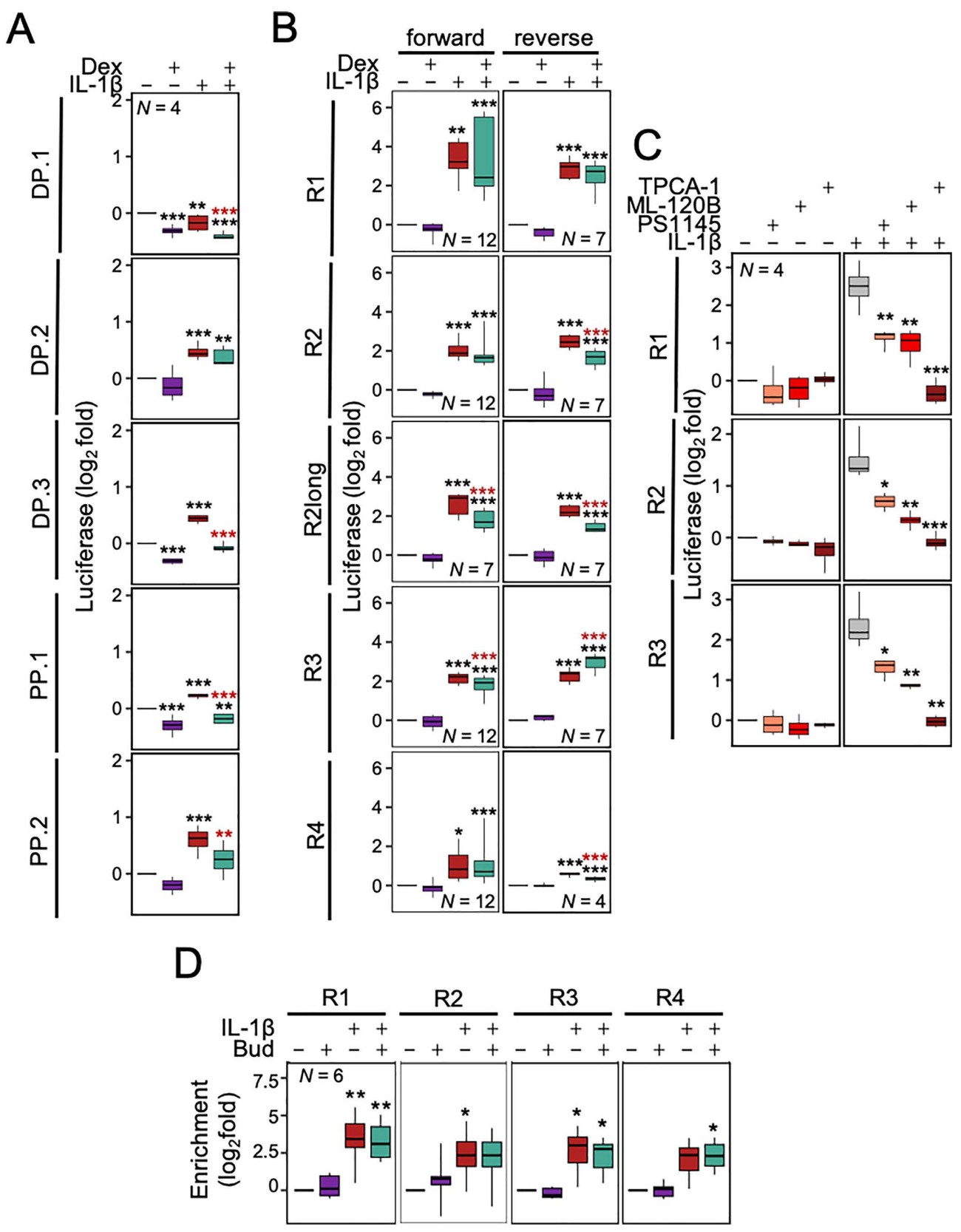

**Fig 8. DNA regions upstream of *IRF1* drive IL-1β-inducible transcription.** (A) A549 cells stably transfected with reporter constructs containing long promoter segments in the forwards orientation as depicted in Fig 7D (DP: Distal Promoter, PP: Proximal Promoter). Reporter cell lines were either not stimulated, or treated with IL-1β (1 ng/ml) and/or dexamethasone (1 µM; Dex) for 6 h prior to luciferase activity determination. (B) A549 cells stably transfected with reporter constructs containing short enhancer segments cloned in both forward and reverse orientations. Reporter cell lines were either not stimulated, or treated with IL-1β (1 ng/ml) and/or dexamethasone (1 µM; Dex) for 6 h prior to luciferase activity determination. (C) A549 cells stably transfected with reporter constructs containing the forward orientation of the short enhancer segments in Fig 7D were incubated with: TPCA-1, ML-120B or PS-1145, each at 30 µM, for 1.5 h prior to treatment with IL-1β (1 ng/ml), as indicated, and harvested after 6 h for luciferase determination. (D) A549 cells were either not-stimulated (NS) or stimulated with budesonide (300 nM, Bud), IL-1B (1 ng/ml) or IL-1B + budesonide for 1 h prior to ChIP-qPCR for p65. PCR primers were designed to span an NF-κB motif (JASPAR, scores ≥ 400) central to a p65 bound region within four p65-bound regions indicated in Fig 7 panel A (R1, R2, R3, and R4). qPCR data were normalized to the geometric mean of three negative control regions that are not occupied by p65. All data from $N$ independent experiments were expressed as (**A-C**) $\log_2$ relative light units (RLU), $\log_2$ fold over not-stimulated, or (**D**) $\log_2$ fold enrichment relative to no treatment control ($\Delta\Delta C_T$) and are shown as box-and-whisker plots. Using fold or $\Delta\Delta C_T$ values, significance was tested by one-way ANOVA with a Tukey's post-hoc test. * $P \le 0.05$, ** $P \le 0.01$, *** $P \le 0.001$ indicates significance relative to (A, B) non-stimulated cells within each reporter construct, or (C) significance relative to naïve cells within the treatment group (non-stimulated or IL-1β stimulated cells), or (D) relative to non-stimulated cells for each region.

IL-1β to 7.7-fold with IL-1β plus dexamethasone. The two R4 regions, which had the weakest p65 enrichment as reporter constructs, showed only 1.5–2.0-fold ($P \le 0.05$) inducibility by IL-1β (Fig 8B). These R4 reporters were unaffected by dexamethasone alone and in the presence of IL-1β there was little effect on the already low reporter drive.

Testing the effect of the IKKβ inhibitors on IL-1β-induced reporter activity showed TPCA-1 to ablate IL-1β-induced activity ($P \le 0.01$) of the R1, R2 and R3 reporters (Fig 8C), effects that were consistent with the repression of IRF1 expression. For R1, ML-120B and PS-1145 reduced IL-1β-induced activity by 66.4% and 63.1%, respectively ($P \le 0.05$) (Fig 8C). Similarly, the activity of R2 and R3 was reduced on average by 57.0% and 45.0% respectively ($P \le 0.05$). Since inducibility in respect of two of the three R4 reporter cell lines was insufficient for reliable quantification, the IKK inhibitors were not tested on these R4 reporter lines.

Consistent with the ChIP-sequencing data, each of the NF-κB-driven regulatory regions, R1-R4, was confirmed as binding p65 by ChIP-qPCR (Fig 8D). Furthermore, there were no significant changes in p65 enrichment at any of the regions in the combination treatment with dexamethasone compared to IL-1β enrichment alone.

## IL-1β and TNFα-induced transcriptional drive from R1, R2, R3 and R4 involves NF-κB motifs

Overall, R1, R2, R3 and, to a much lesser extent, R4, were shown to drive IL-1β-induced transcription, recruit p65 and revealed various centrally located NF-κB motifs. To test the role of each NF-κB motif in driving transcription, site-directed mutagenesis was used to remove each motif from the four short R1-R4 reporter clones. Compared to the wildtype R1 reporter, the ΔS1 construct showed 75.5–79.1% reductions in IL-1β and TNFα induced activity, while the ΔS2 mutant reduced each response by 89.9–90.5% (all $P \le 0.001$) (Fig 9). With the R2 reporter, the ΔS6 deletion reduced inducibility by IL-1β and TNFα by 35.0–45.2%, while the ΔS7 deletion revealed a 51.4–58.1% reduction in each response (all $P \le 0.001$) (Fig 9). For the R3 reporter, the ΔS8 deletion completely ablated IL-1β inducibility by both IL-1β and TNFα (both $P \le 0.001$). Lastly, the low inducibility of the R4 reporter resulted in a more minimal effect with the ΔS10 deletion. Nevertheless, reductions of 14.8% ($P \le 0.01$) and 21.9% ($P \le 0.001$) were apparent for IL-1β- and TNFα-induced reporter activity.

## Effect of inflammatory cytokines and glucocorticoids on expression of IRFs in pHBECs

Messenger RNA-seq was also performed on pHBECs that were grown as submersion or air-liquid interface (ALI) culture and treated with IL-1β (1 ng/ml) and/or either dexamethasone (1 µM) or budesonide (300 nM). Despite initially being quite lowly expressed in pHBECs in submersion (8.82 tpm – 17.1 tpm) and ALI (21.9 tpm – 45.9 tpm), IRF1 expression was dramatically increased by IL-1β treatment in both models (Fig 10A). In submersion pHBECs, the upregulation of IRF1 mRNA by IL-1β peaked at 2 h (21.0 fold, $P \le 0.001$) before declining by 6 h and 24 h (Fig 10B). There was no significant

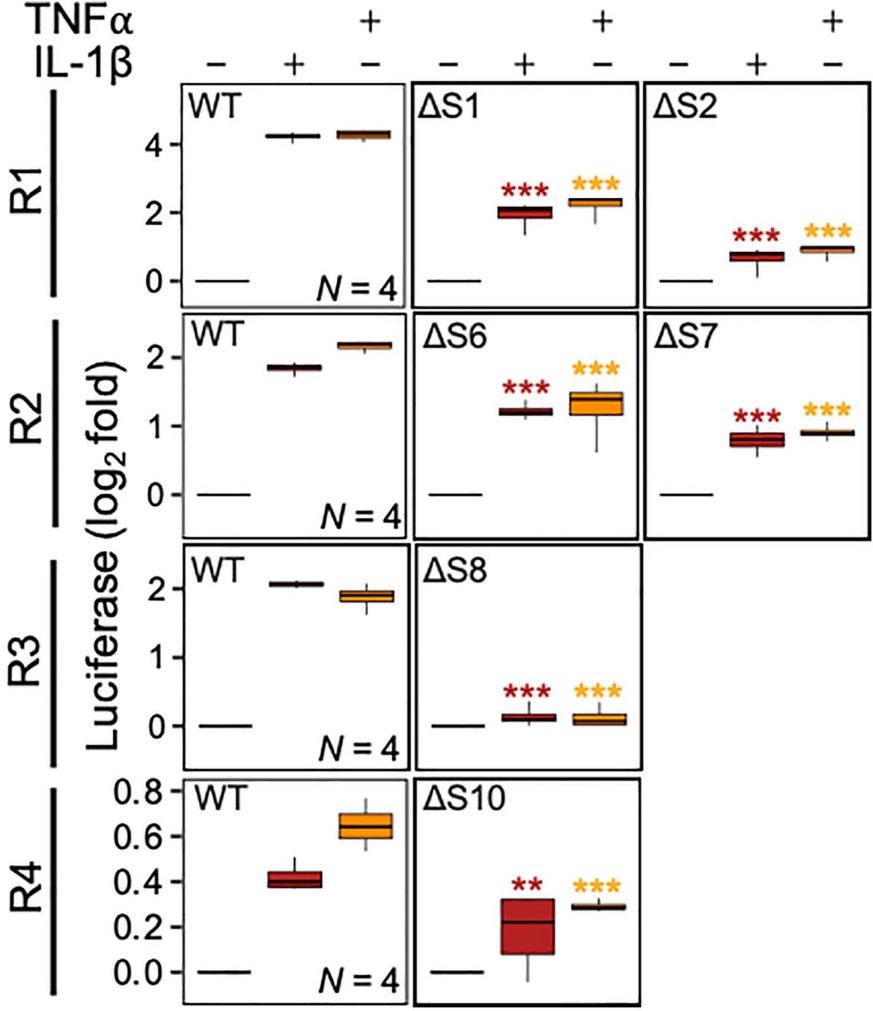

**Fig 9. Roles for NF-κB motifs within the *IRF1* promoter in driving IL-1β and TNFα-inducible transcription.** A549 cells were stably transfected with the R1-R4 forward orientation reporter constructs from Fig 8B and with constructs in which 6 of the numbers NF-κB motifs were deleted (Δ) (location/numbering as shown in Fig 7D). Each batch of reporter lines were generated and treated in parallel. Reporter lines were treated with either IL-1β (1 ng/ml) or TNFα (10 ng/ml) for 6 h prior to harvesting for luciferase determination. Data from *N* independent experiments were expressed as log$_2$ fold over not-stimulated and are shown as box-and-whisker plots. Using fold values, significance was tested by one-way ANOVA with a Tukey's post-hoc test. * $P \leq 0.05$, ** $P \leq 0.01$, *** $P \leq 0.001$ indicates significance relative to the respective wild-type (WT) reporter construct treated with IL-1β (red) or TNFα (yellow).

impact of dexamethasone on basal IRF1 expression nor on IL-1β-upregulated IRF1 at any time point. The ALI pHBECs were treated for 6 h and revealed 5.9-fold ($P \leq 0.05$) upregulation of IRF1 by IL-1β that was unaffected by budesonide co-treatment (**Fig 10A**).

In both submersion and ALI cultured pHBECs, transcripts for IRF2–7 and IRF9 were also detected (**Fig 10B**, **S6 Fig in S1 File**). In submersion culture, IRF6 was detected at the highest levels (103.3–212.7 tpm), followed by IRF3 (53.3–164.9 tpm), with IRFs 9, 5, 2, 7, and 1 ranging from 21.9 down to 3.83 tpm (**Fig 10B**). Similarly, IRF6 was also the most highly expressed (80.3–114.7) in the ALI pHBECs with IRFs 2, 3, 9, 1, 5, 7 all ranging from 93.1 to 14.8 tpm. In both models, IRF4 was very lowly expressed (0.00–0.74 tpm) and IRF8 was undetectable in the absence of stimulation (**Fig 10B**). In the presence of IL-1β, IRF2 mRNA was modestly increased in both the submersion pHBECs at 24 h (1.4-fold, $P \leq 0.05$)

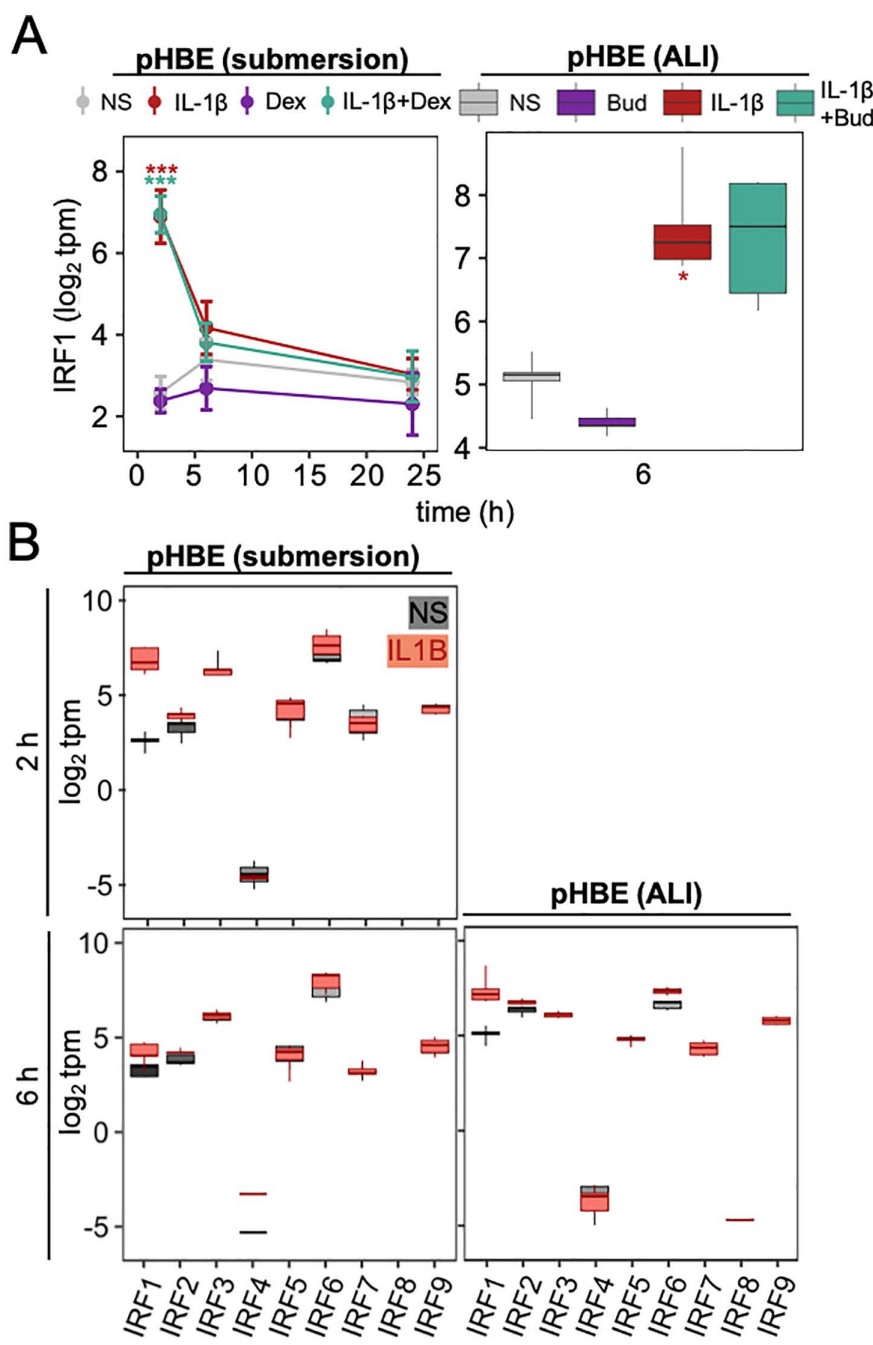

**Fig 10. IRF expression in airway epithelial cells.** Primary human bronchial epithelial (pHBE) cells, grown as submersion culture or at air-liquid interface (ALI), were either not stimulated (NS) or treated with IL-1β (1 ng/ml) and/or dexamethasone (1 μM; Dex) or budesonide (300 nM; Bud) for the indicated times. RNA from 4 individuals (pHBE) was prepared prior to RNA sequencing and results are presented as $log_2$ transcripts/million (tpm). All data are plotted as box-and-whisker plots. In (A), tpm values for IRF1 were used for significance testing by one-way ANOVA with Tukey's post-hoc test * $P \le 0.05$, *** $P \le 0.001$ indicates significance relative to NS.

and the ALI pHBECs at 6 h (1.4-fold, $P \leq 0.001$). Additionally, IRF6 appeared to be induced by IL1β in submersion pHBECs at both 2 h and 6 h (1.6–1.8-fold) and this effect was more robust (1.7 fold, $P \leq 0.001$) in the ALI cultured cells. While IRF8 expression became detectable in the ALI pHBECs, there was little or no effect of IL-1β on the remaining IRFs. In all cases, these patterns of expression were not materially affected by the presence of glucocorticoid (**S6 Fig in S1 File**).

## Discussion

Upregulation of IRF1 expression by IL-1β and TNFα is described in A549 and BEAS-2B cells, along with a similar induction by IL-1β in pHBECs grown in submersion or as highly differentiated pseudostratified cells in ALI culture. In these studies, the consistency of IRF1 upregulation observed between the cell lines and primary cells not only indicates a likely relevance in the context of inflammation, but also suggests that common mechanistic underpinnings may apply across models. Increased IRF1 transcription rate observed 0.5–2 h post-IL-1β treatment is consistent with NF-κB activation, where nuclear translocation and DNA binding occurs within 30–60 minutes of IL-1β or TNFα treatment in A549 cells [41]. Furthermore, since these cytokines not only activate NF-κB in A549 cells, but also in BEAS-2B cells and in pHBECs, NF-κB may represent a common mechanism for IRF1 upregulation across all these AEC models [25,27,41,42]. Since IκBαΔN overexpression, IKKβ inhibitors (TPCA-1, PS-1145, ML-120B), and p65 silencing all reduced IL-1β-induced expression of IRF1, a major role for NF-κB is clearly supported. Notably, TPCA-1 exhibited a considerably greater inhibitory effect relative to either PS-1145 or ML-120B at reducing IRF1 expression. Since IKKα and IKKβ are increasingly recognized as both being necessary for NF-κB activation, the relative lack of IKKβ/IKKα selectivity for TPCA-1, compared to the 1000-fold IKKβ selectivity for ML-120B and PS-1145, may explain these data [29,43,44]. Additionally, four discrete DNA regions upstream of the *IRF1* TSS bound p65, and all drove IL-1β and TNFα inducible transcriptional activity via defined NF-κB motifs, thus direct roles for NF-κB acting at the *IRF1* promoter are indicated.

In analyzing *IRF1* promoter function, cloning of longer (1–3 kb) DNA regions failed to produce reporter constructs with meaningful inducibility, this despite containing p65 binding regions that, when cloned individually, produced marked IL-1β- and TNFα-induced reporter activity. This effect is consistent with shorter distances between cloned enhancers and the TATA/TSS region in reporter plasmids being optimal for transcriptional activity [40]. Such outcomes may relate to variables including; the spacing necessary for optimal interaction between factors binding the enhancer and cofactors or other basal transcriptional factors at the TATA/TSS region, a reduced likelihood of small cloned regions to contain inhibitory motifs, or simply the increased likelihood of disruption of larger fragments on stable transfection [40,45]. These observations underscore the importance of independent unbiased approaches to identify potential enhancer sequences. For example, POL2 is commonly found at active enhancers and was readily apparent at the R1, R2 and R4 p65-binding regions following IL-1β or TNFα treatment of A549 or BEAS-2B cells. Thus, p65 and POL2 recruitment to these regions is consistent with their ability to drive IL-1β- and TNFα-induced transcriptional activity. However, p65 bound R3 and this region was active in reporter assays, yet did not recruit POL2 in either A549 or BEAS-2B cells. Furthermore, strong POL2 enrichment was apparent 3′ to the *IRF1* gene locus and may indicate transcriptional termination, or stalling, for example following rapid cytokine-induced processivity through the *IRF1* gene body, rather than increased transcriptional activity *per se*. Indeed, both poised POL2 at the 5′ start of genes (requiring a signal to initiate transcription) and paused POL2 at the 3′ end of genes (associated with transcriptional termination) are commonly observed, making the interpretation of POL2 presence challenging [37,46]. Alternative to POL2 presence, chromatin modifications, such as H3K27Ac, can mark transcriptionally active regions [34,37,47,48]. In A549 cells, H3K27Ac flanks one or both sides of the upstream R1 and R2 regions, as well as the more proximal R4-containing region just 5′ to the *IRF1* TSS (Fig 7D). While not present at R3, such patterns may indicate prior histone sliding to enable factor access to motifs and are consistent with transcriptional activation at R1, R2 and the R4/TSS regions [34]. Since the H3K27Ac data were only available for unstimulated cells, there remains the possibility that R3 could become activated and show inducible H3K27Ac following IL-1β treatment, as has been reported for other gene enhancers/promoters [49,50]. Nevertheless, when combined with a lack of POL2, it appears that R3,

despite recruiting p65 and, once cloned, being capable of IL-1β-, or TNFα-, induced transcriptional activity, may not play a key role in driving IRF1 expression in the current model. This highlights the importance of using complementary methods to assess roles for the binding regions of specific factors in regulating transcriptional activity. Thus, the identification of novel enhancers should not rely solely on cloning approaches, but also requires appreciation of the native chromatin environment (ChIP sequencing of H3K27Ac) and POL2 presence. Further support for key roles of R1, R2 and the R4/TSS regions also comes from their high levels of genetic conservation as this is a well-established means of identifying functionally important regulatory regions [33]. Thus, NF-κB motifs deleted in the current study showed higher conservation than the surrounding DNA. This was particularly apparent for S2, within R1, and S7, within R2, and such effects are consistent with greater roles for these motifs relative to less conserved motifs (S1 and S6, respectively), as suggested by the deletion analysis. While a block of relatively high conservation at R3 suggests regulatory importance for this region, the S8 NF-κB motif, despite being essential for reporter drive, was less conserved relative to the motifs within R1 or R2. This may indicate a lesser importance for this S8 motif, which given the marginal p65 binding to this region in BEAS-2B cells is also suggestive of a minor role.

The description of direct activation by NF-κB of *IRF1* gene transcription greatly contributes to the understanding of IRF1 transcriptional regulation. This complements studies, primarily in leukocytes, that describe the ability of interferons, including IFNγ, to activate STAT1 which then binds the *IRF1* promoter to induce IRF1 expression [15,51,52]. Such studies identified the γ-interferon-activated sequence, now known as a STAT1 motif, upstream of the *IRF1* TSS (See the STAT1 motif just upstream of TSS in Fig 7D) that appears responsible for IFNγ-induced IRF1 expression [18,19,52]. While prior studies did not establish IRF1 regulation by NF-κB, when taken together with the current study, these collectively explain the upregulation of IRF1 by IFNγ and inflammatory cytokines, including by IL-1β and TNFα, in leukocytes as well as structural cells that include the airway epithelium and smooth muscle [16,53,54].

IL-1β also strongly activated transcription from an "IRF" reporter and this is consistent with induced, and presumably active, IRF1 protein in the nucleus. However, sequence overlap exists between motifs bound by the various IRF proteins, with these all being variations of the 10 base pair motif: 5'GAAANNGAAA 3' [32,55]. Thus, silencing was used to confirm a major role for IRF1 in IL-1β-induced activation of this reporter. While IRF2 was expressed in all the AEC models and presence in the nucleus is consistent with constitutive activity, possibly as a repressor [7,56], we were unable to document a role on IL-1β-induced IRF reporter activity. Similarly, IRF3 expression was high in each AEC model. While IRF3 protein was readily detected in A549 cells, this was unaltered by IL-1β, remained cytoplasmic following IL-1β treatment, and IRF3 silencing did not affect IL-1β-induced IRF reporter activity. Thus, despite no evidence of IRF3 activation by IL-1β being obtained, its constitutive presence suggests that activation, possibly via viral or bacterial components binding to PRRs [10,57], could occur. Likewise, IRF9 expression remained largely cytoplasmic following IL-1β treatment of A549 cells and more modest effects of IRF9 silencing were detected on the IRF reporter. However, while IRF9 expression was upregulated by IL-1β in A549 cells, this was not observed in pHBECs grown at submersion or ALI. As a consequence, the relevance of this effect was unclear in AECs and further analysis was not pursued. Nevertheless, relatively high levels of expression for IRF9, as well as IRF2 and 3, across A549, BEAS-2B and pHBECs, all support possible roles in the context of appropriate stimuli. Similarly, while the inducibility of IRF7 mRNA observed in A549 cells was not apparent in the pHBEC models, expression was at levels that may be functionally relevant. Likewise, expression of IRF6 in the primary cells was greater than IL-1β-induced levels of IRF1 and yet this IRF revealed very much lower expression in both A549 and BEAS-2B cells. Finally, while expression of IRF4 and 8 were consistently low across all the models, expression of IRF5 was also at a level that might have functional correlates. Taken together these data highlight the importance of further studies to interrogate IRF function in suitable AEC models.

Glucocorticoids are the cornerstone of asthma therapy due to their potent anti-inflammatory effects; however, a subset of patients with severe asthma exhibit apparent resistance to these agents, leading to persistent inflammation and worse clinical outcomes [58]. The current study sheds light on the relationship between glucocorticoids and the induction of IRF1

by NF-κB. Transcriptional drive induced by IL-1β, or TNFα, from R1, R2, R3 or R4 involved NF-κB but was only modestly, if at all, affected by glucocorticoid co-treatment. Such data were also consistent with effects on POL2 presence, the lack of glucocorticoid repression on IL-1β-induced transcription rate, and the only modest repressive effects of glucocorticoids on IRF1 expression in A549 and BEAS-2B cells. These data support that IRF1 is integral to glucocorticoid resistance, potentially due to maintenance of IRF1 expression by glucocorticoid-dependent mechanisms [16,23,53]. Thus, while a combination of evolutionary conservation, histone acetylation, POL2 enrichment, binding of p65, and transfection of cloned reporter plasmids allowed the identification of novel regulatory regions that bound NF-κB to drive IRF1 expression, the data reveal how a generic repression of NF-κB by glucocorticoids does not account for the minimal repression of IRF1 by glucocorticoids. Since IRF1 induction by NF-κB was not materially impacted by glucocorticoid, downstream IRF1-dependent gene expression may also be insensitive to glucocorticoids. IRF1 could therefore be explored as a therapeutic target in glucocorticoid resistant diseases such as severe asthma, or high levels of IRF1 expression may act as a biomarker of glucocorticoid insensitive disease [59].

## Materials and methods

### Submersion and air-liquid interface culture of cells

Human pulmonary type 2 A549 cells (CCL-185, American Type Culture Collection (ATCC), Manassas, VA) were cultured in Dulbecco's modified Eagle's medium (DMEM) enriched with 10% fetal bovine serum (FBS) (ThermoFisher Scientific). Primary human bronchial epithelial cells (pHBECs) were grown in PromoCell medium (Sigma-Aldrich) and BEAS-2B cells (CRL-9609; ATCC, Manassas, VA) were grown in DMEM/F12 (ThermoFisher Scientific) also supplemented with 10% FBS. Cells were grown in submersion culture at 37°C in 5% $CO_2$ and passaged when confluent. Prior to all experiments, cells were incubated overnight with serum-free media. Air-liquid interface (ALI) culture of pHBECs was performed as previously described [60]. Cells were seeded into transwell inserts coated with bovine collagen and incubated as above. The apical media was removed, and cultures were fed basally with medium to differentiate cells. After 14 days, cells were washed apically once per week to remove mucus. Highly differentiated ALI cultures were used for experiments ~5 weeks post-transwell seeding.

### Stimuli, drugs and inhibitors

Recombinant human IL-1β (LB-005, R&D Systems) and TNFα (TA-020, R&D Systems) were dissolved in PBS with 0.1% bovine serum albumin (A3059, Sigma-Aldrich). Budesonide (gift from AstraZeneca), dexamethasone (D4902, Sigma-Aldrich), PS-1145 (P6624, Sigma-Aldrich), ML-120B (SML1174, Sigma-Aldrich), TPCA-1 (2559, Tocris) were reconstituted in dimethyl sulfoxide. Final dimethyl sulfoxide concentrations on cells were ≤ 0.1%. G418 (A1720, Sigma-Aldrich) was dissolved in water at 100 mg/ml and filter sterilized.

### RNA isolation, cDNA synthesis SYBR green RT PCR

RNA was extracted using the NucleoSpin RNA extraction kit (MN-740955, D-Mark Biosciences) and 0.5–1 µg was used to synthesize cDNA using a commercial kit (101414, Quantabio). qPCR was carried out using FAST SYBR Green Master Mix (4385618, ThermoFisher Scientific) and analyzed using QuantStudio 3 Real-Time PCR Systems (ThermoFisher Scientific). Amplification conditions were: 95 °C for 20 s, then 40 cycles of 95 °C for 3 s, 60 °C for 30 s. Relative cDNA concentrations were obtained from standard curves generated by serial dilution of cDNA from samples treated with IL-1β and analyzed at the same time as experimental samples. Primers were designed using Primer BLAST (NCIB) and ordered via ThermoFisher Scientific. All primer sequences are provided as supplementary data (**S1 Table in S1 File**).

### RNA-seq

RNA was extracted as above and samples were submitted to the Centre for Health Genomics and Informatics, University of Calgary, for sequencing of polyA-selected mRNA. RNA sequencing libraries, from 4 independent experiments, were prepared using Illumina TruSeq Stranded mRNA Library Prep kits with the poly(A) mRNA magnetic isolation module as

described by the manufacturer. Libraries were validated by D1000 Screen Tape assay on an Agilent 2200 TapeStation system and quantified using Kapa qPCR Library Quantification kits. Libraries were pooled and sequenced across 4 consecutive 75 cycle high-throughput sequencing kits on a NextSeq 500 instrument (Illumina) to generate ~20 million reads per sample. Demultiplexing of the sequencing data was performed using bcl2fastq conversion software and read quality assessed sample using FastQC (Illumina). Good-quality reads were mapped to GRCh38/hg38 reference human transcriptome using kallisto with 100 bootstraps per sample [61]. Normalization and differential expression analysis was performed using the R package, sleuth [62]. Genes with low abundance (<5 estimated counts in at least 90% of all samples) were filtered out before subsequent analysis. Normalized transcripts per million (tpm) was used for graphical presentation of RNA-seq data using the R package, tidyverse. A549 RNA-sequencing data are available in Gene Expression Omnibus (GEO) under accession GSE295743 and raw tpm values for the data used in this article are available upon request.

## Chemiluminescent and fluorescent western blotting

Cells were lysed with 1×Laemmli buffer supplemented with a protease inhibitor cocktail (ThermoFisher Scientific) and β-mercaptoethanol. Alternatively, cells were fractionated with the NE-PER™ Nuclear and Cytoplasmic Extraction Kit (ThermoFisher Scientific) prior to addition of Laemmli, β-mercaptoethanol and protease inhibitors as per manufacturers procedure. After denaturation, proteins in lysates were separated by sodium dodecyl sulfate–polyacrylamide gel electrophoresis using 10% polyacrylamide gels [63]. Proteins were then transferred to a nitrocellulose membrane using a Trans-Blot Turbo Transfer system (Bio-Rad) according to the manufacturer's procedure. After transfer, membranes were blocked with 5% milk in tris-buffered saline with Tween 20 (TBST) prior to incubating with primary antibodies. Membranes were incubated with a horseradish peroxidase-linked secondary antibody, followed by horseradish peroxidase substrate (HRP), for enhanced chemiluminescent visualization with a ChemiDoc MP imaging system (Bio-Rad). Alternatively, membranes were incubated with fluorescently labelled secondary antibodies for fluorescent visualization with the same imaging system. Relative protein expression was quantified by densitometry. Primary antibodies were; IRF1 (D5E4, Cell Signalling), IRF2 (E9S1E, Cell Signalling), IRF3 (D83B9, Cell Signalling), IRF9 (76684S, Cell Signalling), GAPDH (MCA4739, Bio-rad). Fluorescent antibodies were: GAPDH hFAB™ Rhodamine (12004167, Bio-rad), Goat ANTI-RABBIT IgG StarBright™ Blue (12004161, Bio-rad), and Goat anti Mouse IgG (H/L):DyLight®800 (SA5–10176, Bio-rad).

## Adenovirus infection

Cells were grown to ~70% confluency and infected with Ad5-IκBαΔN or Ad5-GFP adenovirus at the indicated MOIs in serum-containing medium, as previously described [27]. After 24 h, the cells were serum-starved prior to treatment.

## siRNA-mediated gene silencing

A549 cells plated at 50–70% confluence were transfected with siRNAs. Pools of 4 non-targeting siRNAs (SI03650325, SI03650318, SI04380467, SI01022064), p65 siRNA (SI02663101, SI05146204, SI00301672, SI02663094), IRF1 siRNA (SI02628080, SI00034104, SI00034090, SI00034083), IRF2 siRNA (SI05120346, SI00034125, SI00034111, SI00034118), IRF3 siRNA (SI05587617, SI03117359, SI03101567, SI02626526), IRF9 siRNA (SI03037650, SI00084378, SI00084371, SI00084364) (all from Qiagen), were mixed with Lipofectamine RNAiMax (13778150, ThermoFisher Scientific) in Opti-MEM (31985070, ThermoFisher Scientific) and then incubated at room temperature for 10 min as per the RNAiMax procedure. Following 48 h incubation of 1 nM siRNA mixtures (or as otherwise indicated), cells were serum starved and treated as indicated.

## Chromatin immunoprecipitation

Chromatin immunoprecipitation (ChIP) was performed as previously described [64]. A549 cells were grown to >80% confluence, serum starved, and treated as indicated. Following treatments, formaldehyde was added to the culture medium

at a final concentration of 1% to cross-link protein-DNA for 10 minutes (PI28906, ThermoFisher Scientific). Cross-linking was halted by incubating with glycine (125 mM) at room temperature for 5 min. The cells were then washed with PBS prior to cytoplasmic and nuclear lysis. Samples were sonicated at 4 °C using a Bioruptor (Diagenode) set to 30 high-power bursts with a 30 s on-off cycle. Protein G magnetic Dynabeads (10004D, ThermoFisher Scientific) were incubated with lysates on roller overnight at 4 °C. Beads were then washed with various buffers as previously described. Crosslinks were reversed, then DNA was purified with a ChIP DNA Clean & Concentrator kit (D5205, Zymo Research) prior to qPCR using Fast SYBR Green Master Mix (ThermoFisher Scientific) as described above. Purified DNA was then analyzed by qPCR, or used to prepare libraries for sequencing (as described below). ChIP-qPCR primers were also designed to amplify the p65-binding region upstream to the regions showing p65-binding within the IRF1 promoter based on the p65 ChIP-seq data in BEAS-2B cells. Relative occupancy at each region was calculated as ΔΔCT after normalization to the geometric mean of $C_T$ values for 3 negative control regions (*OLIG3*, *MYOD1*, and *MYOG*) not predicted to be occupied by p65. All primer sequences are provided as supplementary data (**S1 Table in S1 File**).

### ChIP-Seq

ChIP DNA samples as prepared above, from 2 independent experiments were submitted to the Centre for Health Genomics and Informatics, University of Calgary, for sequencing. Following library preparation (NEB Ultra II kit), samples were subjected to 100-cycle paired-end sequencing (2x50 bp) on Illumina NovaSeq 6000 using NovaSeq SP kit v1.5. Demultiplexing was performed using bcl2fastq conversion software (v2.18.0.12) and read quality was assessed using FastQC (v0.10.1). Good-quality reads were mapped to GRCh38 reference genome using bowtie2 (v2.4.4) and the resultant BAM files were sorted and indexed using samtools. A549 ChIP-sequencing data are available under GEO accession GSE296100 and GSE296101 and links for the data used in this article are available upon request. BEAS-2B ChIP-seq data are available in GEO under accession GSE79803.

### Reporter lines, cloning and site directed mutagenesis

The IRF luciferase reporter was generated using *Kpn*I (R6341, Promega) and *Xba*I (R6181, Promega) to remove the multiple cloning region and luciferase gene from pGL3basic.neo [65] (**S4A Fig in S1 File**). Six copies of an IRF binding site (5′-GGAAGCGAAATGAAATTGACT-3′) upstream of a minimal promoter and the luciferase gene were removed using *Kpn*I and *Xba*I from an IRF1-luc reporter (Panomics) and cloned into the *Kpn*I/*Xba*I cut pGL3basic.neo plasmid backbone to generate an IRF reporter plasmid that was suitable for stable transfection. This was transfected into A549 cells where all G418-resistant cells, representing numerous integration events, were pooled as previously described [66]. Cells were cultured (as above) in the presence of G418 (1 mg/ml) to select for cells containing the reporter plasmid. After 8 h of cytokine stimulation the cells were harvested for luciferase activity determination using a commercial kit (Promega, Madison, WI).

Short enhancer segments (R1-R4) and long promoter regions (DP.1–3, PP.1–2) upstream the *IRF1* gene were PCR amplified from genomic DNA using Platinum SuperFi PCR Master Mix (12358010, ThermoFisher Scientific). PCR primers were designed to contain flanking enzyme sites and, following digestion with *Kpn*I (R3142, New England Biolabs) and *Xho*I (R0146, New England Biolabs), were ligated directly into *Kpn*I/*Xho*I-digested pGL3.TATA.neo vector. Deletion of the NF-κB consensus motifs was performed using the Q5 Site-Directed Mutagenesis Kit (E0554S, New England Biolabs) with primers designed using the online tool NEBaseChanger (https://nebasechanger.neb.com/). All plasmids were stably transfected into pre-confluent A549 cells using Lipofectamine 2000 (11668019, ThermoFisher Scientific) as previously described [28]. All primer sequences are provided as supplementary data (**S2 Table in S1 File**).

### Data presentation and analyses

Significant changes in normalized expression or reporter activity were identified by ANOVA followed by Tukey's Honest Significant Difference post-hoc test (for comparing multiple groups) or Dunnet's post-hoc test where appropriate.

Concentration-response curves were generated using GraphPad Prism 9 software (Graph-Pad Software, San Diego, CA), heatmaps were generated using the R package "*pheatmap*", and all other figures were generated using the R package "*ggplot2*" and modified using the opensource software Inkscape (V.12.2). Data are summarized as line graphs depicting means ± standard error (SE) or as box- and-whiskers plots, where whiskers represent minimum and maximum values and boxes represent lower and upper quartiles and median values. Genomic regions and ChIP-seq data were visualized using the UCSC Genome Browser (https://genome.ucsc.edu).

### Ethics

Non-transplanted normal human lungs obtained from the tissue retrieval service at the International Institute for the Advancement of Medicine (Edison, NJ) were used to isolate primary human bronchial epithelial cells (pHBECs). Family members of donors gave consent for tissues to be used for research. The donor cells used in this study were collected starting with the first pHBECs frozen down on 28-05-2010, and ending with the last pHBECs on 13-11-2016. Archived donor cells were retrospectively used to prepare samples for RNA-sequencing starting on 14-03-2019, and ending on 31-05-2019. This data was accessed for the current study to prepare **Fig 10** up until 05-22-2025, over a period for which the current ethics approval applies. No personal identifying information was provided, and local ethics approval (26-03-2004 to 26-03-2026) was granted via the Conjoint Health Research Ethics Board of the University of Calgary (Historical Study ID: REB04–17694, Current Study ID: REB15–0336). Local ethics approval was also obtained for the use of epithelial cell lines (Study ID: REB19–1574).

### Supporting information

**S1 File. Supplementary figures, figure legends, and tables.** This file contains supplementary figures and legends 1–6, and supplementary tables 1 and 2 which contain the oligonucleotide sequences used in this study.
(PDF)

**S2 File. Uncropped western blot images.** The original, uncropped images for each representative blot are shown with molecular weight marker included. Any lanes not included in the final figure have been marked with an "X" above each lane.
(PDF)

### Author contributions

**Conceptualization:** Amandah Necker-Brown, Robert Newton.

**Data curation:** Amandah Necker-Brown, Sarah K. Sasse, Robert Newton.

**Formal analysis:** Amandah Necker-Brown, Mahmoud M. Mostafa, Sarah K. Sasse, Anthony N. Gerber, Robert Newton.

**Funding acquisition:** Robert Newton.

**Investigation:** Amandah Necker-Brown, Andrei Georgescu, Andrew J. Thorne, Priyanka Chandramohan, Cora Kooi, Keerthana Kalyanaraman, Alex Gao, Akanksha Bansal, Robert Newton.

**Methodology:** Amandah Necker-Brown, Robert Newton.

**Project administration:** Robert Newton.

**Resources:** Anthony N. Gerber, Richard Leigh, Robert Newton.

**Supervision:** Anthony N. Gerber, Richard Leigh, Robert Newton.

**Validation:** Amandah Necker-Brown.

**Visualization:** Amandah Necker-Brown, Mahmoud M. Mostafa, Robert Newton.

**Writing – original draft:** Amandah Necker-Brown, Robert Newton.

**Writing – review & editing:** Amandah Necker-Brown, Mahmoud M. Mostafa, Andrei Georgescu, Andrew J. Thorne, Cora Kooi, Keerthana Kalyanaraman, Sarah K. Sasse, Richard Leigh, Robert Newton.

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
