## [Decision Letter · Decision Letter 0]

9 Sep 2025

Dear Dr. Newton,

Thank you for submitting your manuscript to PLOS ONE. After careful consideration, we feel that it has merit but does not fully meet PLOS ONE’s publication criteria as it currently stands. Therefore, we invite you to submit a revised version of the manuscript that addresses the points raised during the review process.

In your revised manuscript, please extensively work on the abstract. I recommend revising the abstract to improve its logical flow and thematic coherence. The current version is difficult to follow and unlikely to engage a broad scientific audience. As Reviewer #2 suggested, some of the data presentation needs to be reformatted and with suitable statistic analysis. 

We look forward to receiving your revised manuscript.

Kind regards,

Helene Minyi Liu, Ph.D.

Academic Editor

PLOS ONE

Journal Requirements:

“This work was supported by: Grants from the Canadian Institutes of Health Research (CIHR) (funding reference numbers: PJT 156310 (RN) & 180480 (RN)) and Natural Sciences and Engineering Research Council of Canada (NSERC) discovery grants [RGPIN-2016-04549 & RGPIN-2023-03763] (RN); studentships (ANB): Alberta Graduate Excellence Scholarships (2020-2025).”

“This work was supported by: Grants from the Canadian Institutes of Health Research (CIHR) (funding reference numbers: PJT 156310 (RN) & 180480 (RN)) and Natural Sciences and Engineering Research Council of Canada (NSERC) discovery grants [RGPIN-2016-04549 & RGPIN-2023-03763] (RN); studentships (ANB): Alberta Graduate Excellence Scholarships (2020-2025).”

5. We noted in your submission details that a portion of your manuscript may have been presented or published elsewhere. [A549 and BEAS-2B ChIP-sequencing, and A549 RNA-sequencing datasets are published (BEAS-2B) or under review for publication (A549), but no specific data/graphs for IRFs used in this paper are included. All datasets are available in GEO.] Please clarify whether this [conference proceeding or publication] was peer-reviewed and formally published. If this work was previously peer-reviewed and published, in the cover letter please provide the reason that this work does not constitute dual publication and should be included in the current manuscript.

6. We note that the grant information you provided in the ‘Funding Information’ and ‘Financial Disclosure’ sections do not match.

7. Thank you for stating the following in the Acknowledgments Section of your manuscript:

“This work was supported by: Grants from the Canadian Institutes of Health Research (CIHR) (funding reference numbers: PJT 156310 (RN) & 180480 (RN)) and Natural Sciences and Engineering Research Council of Canada (NSERC) discovery grants [RGPIN-2016-04549 & RGPIN-2023-03763] (RN); studentships (ANB): Alberta Graduate Excellence Scholarships (2020-2025). All authors declare no competing interests with the contents of this article.”

“This work was supported by: Grants from the Canadian Institutes of Health Research (CIHR) (funding reference numbers: PJT 156310 (RN) & 180480 (RN)) and Natural Sciences and Engineering Research Council of Canada (NSERC) discovery grants [RGPIN-2016-04549 & RGPIN-2023-03763] (RN); studentships (ANB): Alberta Graduate Excellence Scholarships (2020-2025).”

8. PLOS ONE now requires that authors provide the original uncropped and unadjusted images underlying all blot or gel results reported in a submission’s figures or Supporting Information files. This policy and the journal’s other requirements for blot/gel reporting and figure preparation are described in detail at https://journals.plos.org/plosone/s/figures#loc-blot-and-gel-reporting-requirements and https://journals.plos.org/plosone/s/figures#loc-preparing-figures-from-image-files. When you submit your revised manuscript, please ensure that your figures adhere fully to these guidelines and provide the original underlying images for all blot or gel data reported in your submission. See the following link for instructions on providing the original image data: https://journals.plos.org/plosone/s/figures#loc-original-images-for-blots-and-gels.  

9. We note that you have included the phrase “data not shown” in your manuscript. Unfortunately, this does not meet our data sharing requirements. PLOS does not permit references to inaccessible data. We require that authors provide all relevant data within the paper, Supporting Information files, or in an acceptable, public repository. Please add a citation to support this phrase or upload the data that corresponds with these findings to a stable repository (such as Figshare or Dryad) and provide and URLs, DOIs, or accession numbers that may be used to access these data. Or, if the data are not a core part of the research being presented in your study, we ask that you remove the phrase that refers to these data.

10. Your ethics statement should only appear in the Methods section of your manuscript. If your ethics statement is written in any section besides the Methods, please move it to the Methods section and delete it from any other section. Please ensure that your ethics statement is included in your manuscript, as the ethics statement entered into the online submission form will not be published alongside your manuscript.

11. Please include captions for your Supporting Information files at the end of your manuscript, and update any in-text citations to match accordingly. Please see our Supporting Information guidelines for more information: http://journals.plos.org/plosone/s/supporting-information.

12.If the reviewer comments include a recommendation to cite specific previously published works, please review and evaluate these publications to determine whether they are relevant and should be cited. There is no requirement to cite these works unless the editor has indicated otherwise. 

Reviewers' comments:

Reviewer's Responses to Questions

**Comments to the Author**

1. Is the manuscript technically sound, and do the data support the conclusions?

Reviewer #1: Yes

Reviewer #2: Yes

2. Has the statistical analysis been performed appropriately and rigorously?

Reviewer #1: Yes

Reviewer #2: Yes

3. Have the authors made all data underlying the findings in their manuscript fully available?

Reviewer #1: Yes

Reviewer #2: Yes

4. Is the manuscript presented in an intelligible fashion and written in standard English?

Reviewer #1: Yes

Reviewer #2: Yes

Reviewer #1: This manuscript examines the regulation of IRFs in human pulmonary epithelial cells, with a particular emphasis on the NFκB–dependent induction of IRF1 by IL1β and TNFα. The authors use a combination of cell lines (A549 and BEAS-2B) and primary human bronchial epithelial cells, applying multiple complementary approaches, including mRNA-seq, ChIP-seq, siRNA silencing, pharmacologic inhibition, and reporter assays, to dissect the underlying mechanisms.

The data convincingly demonstrate that IRF1 is robustly induced in an NFκB-dependent manner, with several upstream enhancer regions contributing to its transcriptional control. Notably, IRF1 induction appears largely resistant to glucocorticoid suppression, a finding with potential implications for glucocorticoid resistant airway inflammation and severe asthma.

Overall, this is a technically rigorous, well controlled, and carefully executed study that makes an original contribution to the field. The conclusions are well supported by the data, the manuscript is clearly written, and all required ethics approvals and data availability statements are in place.

I have only one minor suggestion:

The abstract is difficult to follow in its current form. It should be revised for clarity and improved flow so that the key findings are more easily understood.

Reviewer #2: During inflammation, cytokines such as IL-1β and TNF⍺ activate signaling cascades through their respective receptors, leading to the activation of transcription factors like NF-κB and interferon regulatory factors (IRFs). NF-κB is activated through the phosphorylation and degradation of IκBα, allowing NF-κB to translocate to the nucleus and drive inflammatory gene expression. Although IRFs are primarily known for their role in antiviral and interferon responses, emerging evidence suggests they may also be activated through NF-κB-associated pathways, though this remains underexplored. Airway epithelial cells (AECs), as the frontline defenders against inhaled pathogens and irritants, play a central role in lung inflammation and are key targets of inhaled glucocorticoids used to manage asthma. However, the regulation of IRFs—particularly IRF1—in AECs is not well understood. IRF1 is inducible by cytokines and may regulate inflammatory mediators such as CXCL10, which has been linked to severe asthma. This study investigates how IL-1β and TNF⍺ influence IRF1 expression and inflammatory signaling in AECs, and how these responses are modulated by glucocorticoids like dexamethasone and budesonide. There are some comments as the follows:

1. In some graphs, such as Fig 1A, Fig 10A, Supplementary Fig 1C, and Supplementary Fig 6A, the time intervals in the x-axis are different, so they are not suitable for line chart.

2. In Fig 3B and 3C, Fig 4, Fig 6A and Fig 9, the statistical results of some charts need to be fully labeled and presented. For example, in Fig 3, some are labeled with “ns”, but it’s confused whether non-labeled are all “ns”.

3. In Fig 3 and Fig 9, should TNF⍺ treatment groups be labeled as yellow or orange?

4. In line 99, “This revealed transcripts for all nine IRFs under basal conditions…." should be “These revealed transcripts……".

**Do you want your identity to be public for this peer review?** For information about this choice, including consent withdrawal, please see our Privacy Policy

Reviewer #1: No

Reviewer #2: No

---

## [Author Response · Author response to Decision Letter 1]

23 Sep 2025

Responses to the reviewer and editor are contained within the attached document "Response to Reviewers PONE-D-25-37763"

---

## [Editor Report · Decision Letter 1]

6 Oct 2025

Inflammatory cytokines promote interferon regulatory factor (IRF) transcriptional activity in human pulmonary epithelial cells through the induction of IRF1 by nuclear factor-κB

PONE-D-25-37763R1

Dear Dr. Newton,

We’re pleased to inform you that your manuscript has been judged scientifically suitable for publication and will be formally accepted for publication once it meets all outstanding technical requirements.

Kind regards,

Helene Minyi Liu, Ph.D.

Academic Editor

PLOS ONE
---

## [Editor Report · Acceptance letter]

PONE-D-25-37763R1

PLOS ONE

Dear Dr. Newton,

I'm pleased to inform you that your manuscript has been deemed suitable for publication in PLOS ONE. Congratulations! Your manuscript is now being handed over to our production team.

Kind regards,

on behalf of

Dr. Helene Minyi Liu

Academic Editor

PLOS ONE